# ZHX2 emerges as a negative regulator of mitochondrial oxidative phosphorylation during acute liver injury

Yankun Zhang[1], Yuchen Fan[2], Huili Hu [3], Xiaohui Zhang[3], Zehua Wang[1], Zhuanchang Wu[1], Liyuan Wang[1], Xiangguo Yu[1], Xiaojia Song[1], Peng Xiang[1], Xiaodong Zhang[1], Tixiao Wang[1], Siyu Tan[1], Chunyang Li [1,4], Lifen Gao [1], Xiaohong Liang [1], Shuijie Li[5], Nailin Li [6], Xuetian Yue [1,7] ✉ & Chunhong Ma [1] ✉

Mitochondria dysfunction contributes to acute liver injuries, and mitochondrial regulators, such as PGC-1α and MCJ, affect liver regeneration. Therefore, identification of mitochondrial modulators may pave the way for developing therapeutic strategies. Here, ZHX2 is identified as a mitochondrial regulator during acute liver injury. ZHX2 both transcriptionally inhibits expression of several mitochondrial electron transport chain genes and decreases PGC-1α stability, leading to reduction of mitochondrial mass and OXPHOS. Loss of *Zhx2* promotes liver recovery by increasing mitochondrial OXPHOS in mice with partial hepatectomy or CCl4-induced liver injury, and inhibition of PGC-1α or electron transport chain abolishes these effects. Notably, ZHX2 expression is higher in liver tissues from patients with drug-induced liver injury and is negatively correlated with mitochondrial mass marker TOM20. Delivery of shRNA targeting *Zhx2* effectively protects mice from CCl4-induced liver injury. Together, our data clarify ZHX2 as a negative regulator of mitochondrial OXPHOS and a potential target for developing strategies for improving liver recovery after acute injuries.

Liver is the largest internal organ of the human body and is responsible for metabolism, immunity, digestion, detoxification, and protein synthesis, and is characterized by robust regenerative capacity in response to injury[1,2]. Mitochondria act as the critical metabolic and signaling hubs to maintain liver homeostasis, flexibility, and survival[3]. Abnormal mitochondrial function has been reported to not only trigger the onset of various liver diseases, but also contribute to acute liver injury and liver failure caused by infection, toxin, and drug abuse[4–8]. Hepatic mitochondrial oxidative capacity varies broadly across the spectrum of obesity and nonalcoholic fatty liver diseases (NAFLD)[7], and studies on the contributions of altered mitochondria in liver metabolic diseases are controversial. For instance, lifestyle modifications and drugs that are able to enhance mitochondrial function are successful in improving NAFLD and NASH[9]. On the contrary, loss of mitochondrial OXPHOS could protect against diet-induced steatosis and NASH progression[10]. Nevertheless, controlling mitochondrial dysfunction might provide a promising strategy forward to the treatment of liver diseases, especially for acute liver injury.

[1]Key Laboratory for Experimental Teratology of Ministry of Education, School of Basic Medical Sciences, Qilu Hospital, Cheeloo Medical College of Shandong University, Jinan, China. [2]Department of Hepatology, Qilu Hospital of Shandong University, Jinan, China. [3]Institute of Molecular Medicine and Genetics, School of Basic Medical Sciences, Shandong University, Jinan, China. [4]Department of Histology and Embryology, School of Basic Medical Sciences, Shandong University, Jinan, China. [5]College of Pharmacy, Harbin Medical University, Harbin, China. [6]Department of Medicine-Solna, Cardiovascular Medicine Unit, Karolinska Institute, Stockholm, Sweden. [7]Department of Cell Biology, School of Basic Medical Sciences, Cheeloo College of Medicine, Shandong University, Jinan, China. ✉e-mail: yuexu@sdu.edu.cn; machunhong@sdu.edu.cn

Emerging evidence demonstrates that mitochondria play a central role in liver regeneration during acute liver injuries[11,12]. After hepatectomy, mitochondria support hepatocyte proliferation by producing ATP through OXPHOS to meet the bioenergetic demands of hepatocytes[13]. Improvement of mitochondrial biogenesis increases mitochondrial OXPHOS to promote liver regeneration after 2/3 partial hepatectomy (PHx)[14]. However, currently limited number of mitochondrial function regulators has been identified in liver injuries. Peroxisome proliferator-activated receptor coactivator-1α (PGC-1α) is essential for the upregulation of electron transport chain (ETC) genes, activation of respiratory chain, increase of mitochondrial mass, and the augmentation of mitochondrial respiratory capacity[15]. TRPM8 (transient receptor potential melastatin 8) enhances metabolism and promotes hepatocytes proliferation in mice after hepatectomy[16]. Methylation-controlled J protein (MCJ) is a distinct co-chaperone that localizes at the mitochondrial inner membrane[17]. The absence of MCJ enhances mitochondrial activity to promote liver regeneration[4,18]. In addition, mutation of mitochondrial transcription factor A (TFAM), another well-known mitochondrial modulator, causes neonatal liver failure associated with mtDNA depletion[19]. Knockout of mitochondrial topoisomerase I (Top1mt) interrupts the biogenesis of mitochondrial and declines the capacity of liver regeneration[20]. Therefore, targeting mitochondrial regulation seems to be an appropriate strategy to improve repair of liver injury and there is an urgent need to identify mitochondrial regulators.

Transcription factor Zinc-finger and homeoboxes 2 (ZHX2) has been identified as a critical regulator of liver postnatal gene expression, cell proliferation, and hepatic lipid hemostasis[21–25]. Importantly, loss of Zhx2 accounts partially for high-fat diet-induced lipid accumulation and liver damage[26]. Further studies demonstrated that ZHX2 inhibits exogenous lipid uptake and de novo lipid synthesis[27,28], both biological processes are closely associated with ATP production from mitochondria. However, whether ZHX2 plays a role in mitochondrial regulation during liver injury and repair is largely unexplored. Here, we show that hepatocyte-specific knockout of Zhx2 enhances mitochondrial OXPHOS and promotes liver recovery after acute injury. Mechanistically, ZHX2 represses expression of mitochondrial ETC genes through PGC-1α-dependent and independent manner to inhibit mitochondrial OXPHOS and reduce mitochondrial mass. Clinical data verify the negative correlation of ZHX2 expression with mitochondrial mass in patients with drug-induced liver injury (DILI). These findings demonstrate that ZHX2 is a regulator of mitochondria function that contributes to the repair of liver injury and may serve as a drug target for acute liver injury with mitochondrial dysfunction.

## Results

### ZHX2 decreases hepatic mitochondrial biogenesis and OXPHOS

To assess the potential involvement of ZHX2 in hepatic mitochondrial regulation, we first performed bioinformatics analysis using a published dataset with ZHX2 manipulation[29]. Gene set enrichment analysis (GSEA) demonstrated the significant enrichment of the gene sets encoding OXPHOS and ETC OXPHOS system in control cells relative to ZHX2-overexpressing L02 cells (Supplementary Fig. 1a). Furthermore, we performed GSEA in acetaminophen-induced acute liver failure (ALF) cohort (GSE74000) using median of ZHX2 expression levels as a cutoff[30]. As shown in Supplementary Fig. 1b, OXPHOS-related gene sets were enriched in low ZHX2 groups from ALF cohorts. Together, bioinformatics analyses indicate that ZHX2 is associated with mitochondrial function.

To confirm the role of ZHX2 in mitochondrial regulation, mitochondrial biogenesis was examined in HCC cell line Huh7 with ZHX2 manipulation (Supplementary Fig. 1c). As shown in Fig. 1a, b, overexpression of ZHX2 in Huh7 cells decreased mitochondrial mass, as evidenced by lower copy number of mtDNA, weaker intensities of Mito Tracker deep red, less total mitochondrial area as well as fewer

mitochondrial filamentous network. In accordance, knockdown of ZHX2 led to augmented mtDNA copy number and higher levels of Mito Tracker deep red in Huh7 cells (Supplementary Fig. 1d). Importantly, ZHX2 greatly inhibited mitochondrial OXPHOS, presented as lower JC-1 aggregates in ZHX2-overexpressing Huh7 cells and higher JC-1 aggregates in ZHX2-knockdown Huh7 cells (Fig. 1c and Supplementary Fig. 1e). Consistently, ZHX2 overexpression reduced oxygen consumption rate (OCR) (Fig. 1d), while ZHX2 silence promoted mitochondrial OXPHOS (Supplementary Fig. 1f). Subsequently, ZHX2 decreased ATP generation, indicated as lower ATP levels, reduced ratios of ATP/AMP and ATP/ADP, and higher AMP levels in ZHX2-overexpressing Huh7 cells than those of control (Fig. 1e). On the contrary, ZHX2 knockdown increased ATP generation (Supplementary Fig. 1g).

To further validate the above results, both human hepatic organoids and murine primary hepatocytes were included. ZHX2 was successfully knocked down in human hepatic organoids (Supplementary Fig. 1h). Importantly, knockdown of ZHX2 increased mitochondrial mass and ATP generation in human hepatic organoids (Fig. 1f). Consistently, ZHX2 knockout markedly increased mitochondrial mass, enhanced mitochondrial membrane potential, and promoted ATP production in primary hepatocytes isolated from hepatocyte-specific Zhx2 knockout mice (Zhx2-KO$^{hep}$) (Fig. 1g and Supplementary Fig. 1i). Above all, these data demonstrate that ZHX2 inhibits mitochondrial biogenesis and OXPHOS in hepatocytes.

### ZHX2 deficiency accelerates liver repair after acute injury

Since mitochondria play important roles in the repair of damaged liver and ZHX2 regulates mitochondrial function, we hypothesized that ZHX2 might involve in liver recovery from damage. To address this, 2/3 partial hepatectomy (PHx), the widely used mice model of acute liver injury, was performed with Zhx2-KO$^{hep}$ mice and their littermate controls (Zhx2-WT). As shown in Supplementary Fig. 2a, Zhx2 mRNA levels were largely reduced in mice liver at 24 h after damage. Although the initial body weight, liver weight, and liver/body weight ratios were comparable between Zhx2-KO$^{hep}$ and Zhx2-WT mice (Supplementary Fig. 2b), the liver/body weight ratios following 2/3 PHx were recovered faster in Zhx2-KO$^{hep}$ mice than that in Zhx2-WT mice, and the liver mass of Zhx2-KO$^{hep}$ mice was larger than that of Zhx2-WT mice at 96 h after 2/3 PHx (Fig. 2a). In line with the rapid liver mass restoration, we observed an earlier induction of Cyclin D1 at 12 h post 2/3 PHx in Zhx2-KO$^{hep}$ mice and this increase of Cyclin D1 lasted until 48 h after PHx in Zhx2-KO$^{hep}$ mice. In agreement, enhanced expression of Cyclin A2, Cyclin B1, Cyclin E1, and proliferating cell nuclear antigen (PCNA) at both mRNA and protein levels were observed at different time points post 2/3 PHx in Zhx2-KO$^{hep}$ mice (Fig. 2b and Supplementary Fig. 2c). Meanwhile, the numbers of BrdU-positive cells and mitotic cells in the livers of Zhx2-KO$^{hep}$ mice were higher than those of Zhx2-WT mice (Fig. 2c and Supplementary Fig. 2d). As expected, the livers of both groups recovered almost completely at 7 days after PHx. We did not detect any significant differences of ALT, AST and H&E staining between Zhx2-KO$^{hep}$ and Zhx2-WT mice (Supplementary Fig. 2e, f). Whereas liver/body weight ratios and liver cell proliferation had no difference between Zhx2-KO$^{hep}$ and Zhx2-WT mice after sham operation (Supplementary Fig. 2g, h).

To test whether ZHX2-mediated liver recovery is specific to 2/3 PHx, carbon tetrachloride (CCl4) exposure was performed on Zhx2-KO$^{hep}$ and Zhx2-WT mice to mimic chemical-induced liver injury. As expected, Zhx2 deficiency significantly increased the expression of cell-cycle-related genes (Cyclin D1, Cyclin A2, Cyclin B1, and Cyclin E1) and PCNA at 48 and 72 h after CCl4 exposure (Fig. 2d and Supplementary Fig. 2i). Notably, loss of Zhx2 decreased the cell death and liver damage caused by CCl4, as assessed by reduced serum levels of ALT and AST, decreased necrotic areas, and lower number of TUNEL-positive cells (Fig. 2e–g). Taken together, these results demonstrate

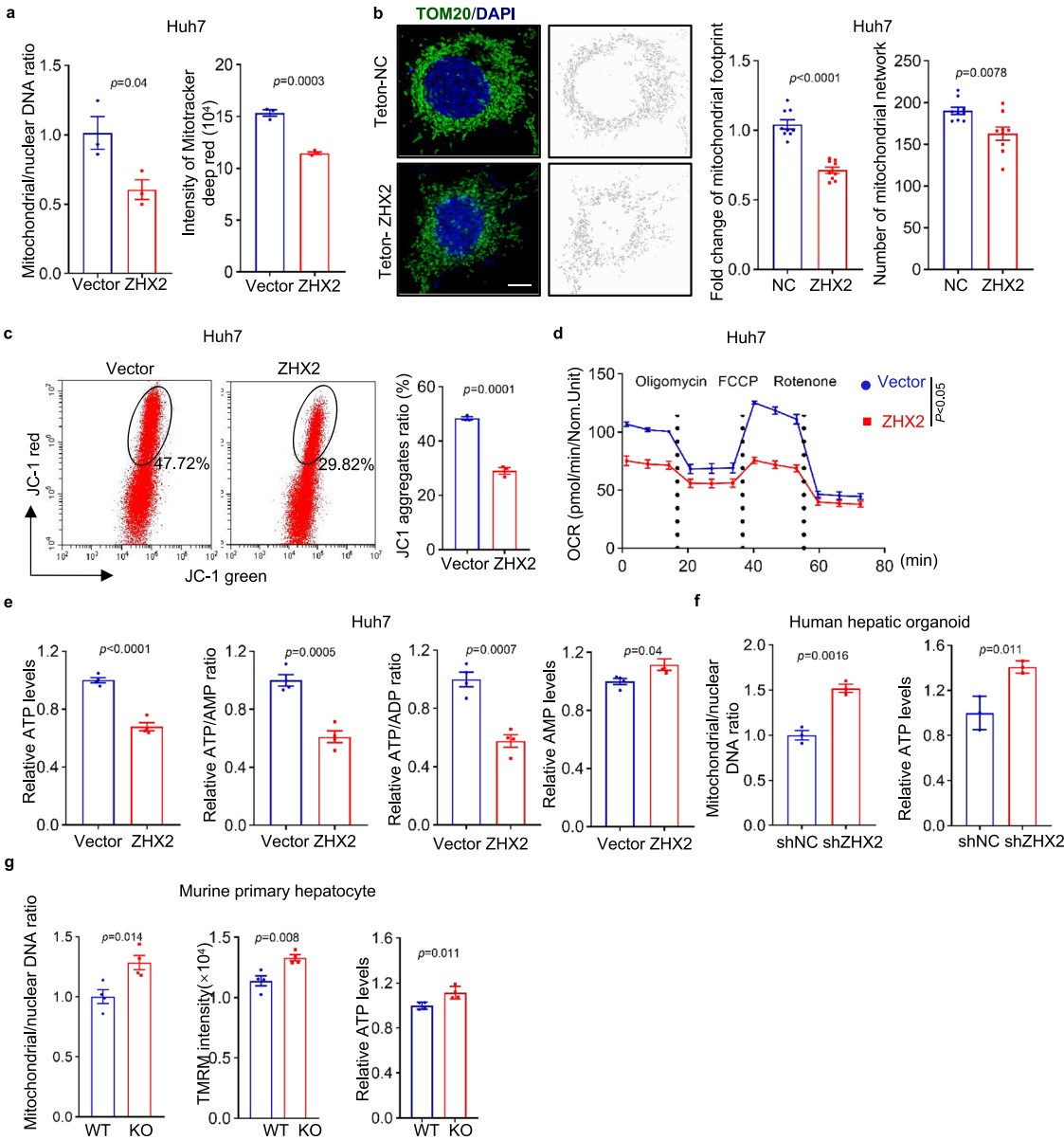

**Fig. 1 | Interruption of ZHX2 enhances hepatic mitochondrial biogenesis and OXPHOS in vitro and in vivo. a** The copy number of mtDNA (left) and intensity of Mito tracker deep red (right) in Huh7 cells with ZHX2 overexpression were determined by qPCR and flow cytometry, respectively. Representative data are presented as mean ± sd (two-tailed Student's *t*-test. *n* = 3 biologically independent samples). **b** Mitochondrial morphology in Huh7 cells with or without ZHX2 overexpression was observed by super-resolution microscopy. Representative images depicting TOM20 (green), DAPI (blue) and mitochondrial network model were presented on the right panel. Quantification of total mitochondrial area (footprint) and mitochondrial networks are shown on the middle and right panel, respectively. Scale bar: 5 μm. Representative data are represented as mean ± sd (two-tailed Student's *t*-test. *n* = 9 cells). **c–e** JC-1 aggregates (**c**), OCR (**d**) and ATP, AMP levels,

ATP/AMP and ATP/ADP ratios (**e**) in Huh7 cells with ZHX2 overexpression were examined by the assay kits. Representative data are presented as mean ± sd (two-tailed Student's *t*-test. **c**: *n* = 3; and **d** and **e**: *n* = 4. *n* represents biologically independent samples). **f** The copy number of mtDNA and intracellular ATP levels were measured in human hepatic organoids transfected with lentivirus-mediated shRNA against ZHX2 (LV-shZHX2) or control shRNA (LV-shNC) with ATP level assay kit. Representative data are presented as mean ± sd (two-tailed Student's *t*-test. *n* = 3 biologically replicates experiments). **g** The copy number of mtDNA, intensity of TMRM and ATP levels in primary hepatocytes from *Zhx2*-WT and *Zhx2*-KO[hep] mice were determined. Representative data are presented as mean ± sd (two-tailed Student's *t*-test. *n* = 4 biologically independent samples).

that hepatic *Zhx2* deficiency promotes liver repair from acute injuries induced either by mechanical or chemical factors.

### *Zhx2* deficiency augments hepatic OXPHOS during acute injury

To decipher the underlying mechanism for liver recovery from an acute injury, we performed multi-omics with liver tissues from *Zhx2*-KO[hep] and *Zhx2*-WT mice at 48 h after 2/3 PHx (Supplementary Fig. 3a). RNA-seq identified a total of 1091 differentially expressed genes (DEGs) (fold change ≥ 2.00, adjusted *p* value ≤ 0.05) (Supplementary Fig. 3b).

Gene Set Variation Analysis (GSVA) revealed multiple upregulated gene sets in *Zhx2*-KO[hep] mice, including E2F targets and G2M checkpoint (Supplementary Fig. 3c), two known signatures related with damaged liver recovery[31,32]. Notably, metabolism-related gene sets were upregulated in *Zhx2*-KO[hep] mice, of which OXPHOS was the most significantly upregulated gene set (Supplementary Fig. 3c). GSEA further showed the enrichment of mitochondrial related gene sets, including OXPHOS gene set (Fig. 3a). Consistently, quantitative proteomics identified a total of 644 differentially expressed proteins (fold

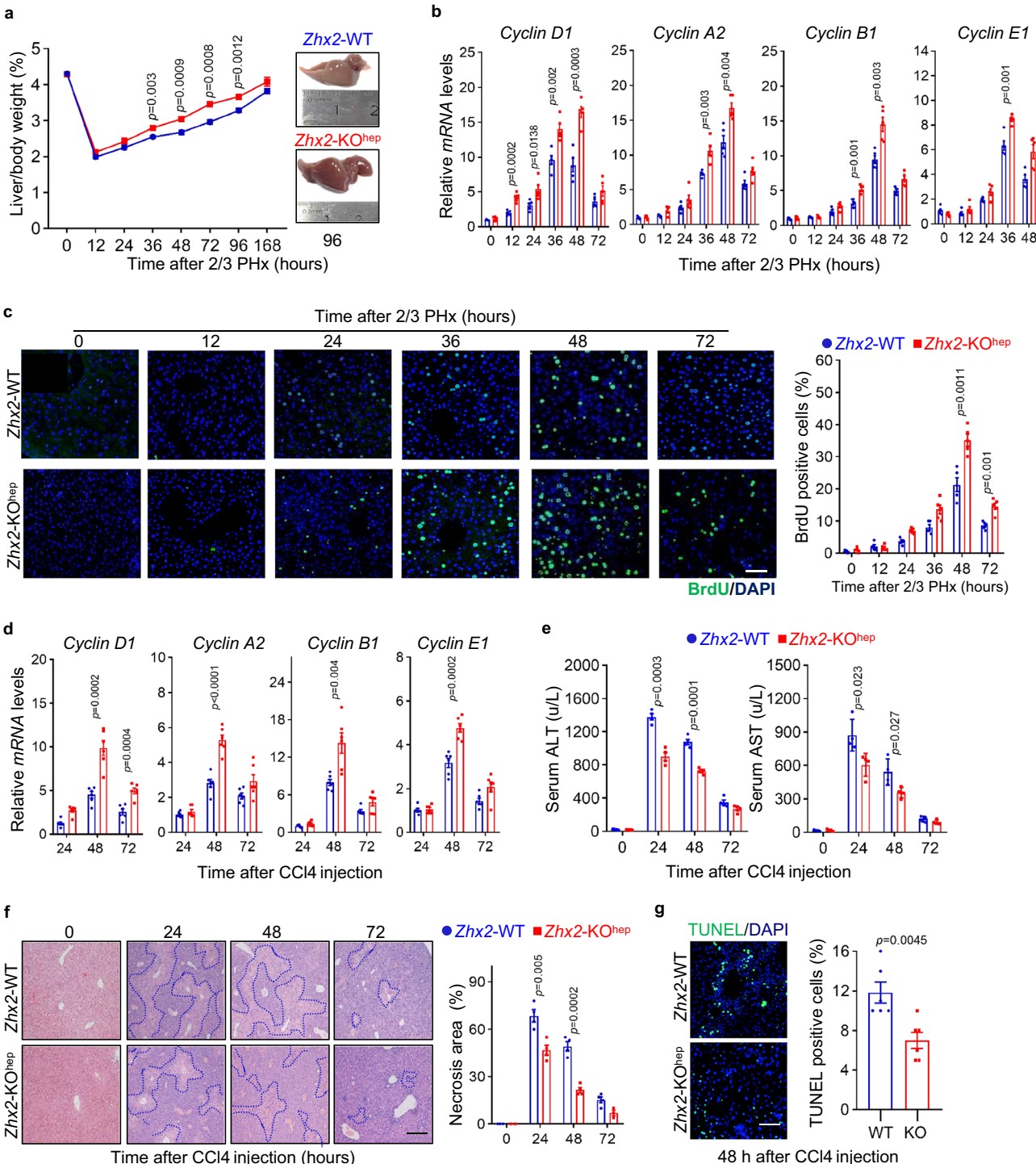

**Fig. 2 | Liver repair is enhanced in *Zhx2*-KO^hep mice after 2/3 PHx and CCl4 administration. a** Liver/body weight ratios of *Zhx2*-WT and *Zhx2*-KO^hep mice at indicated time points after 2/3 PHx are presented on the left panel. Right panel displays the representative images of liver mass at 96 h after 2/3 PHx. Data are presented as mean ± s.e.m. (two-tailed Student's *t*-test. *n* = 5 mice per group). **b** The mRNA levels of *Cyclin D1, Cyclin A2, Cyclin B1* and *Cyclin E1* were accessed in *Zhx2*-WT and *Zhx2*-KO^hep mice at indicated time points after 2/3 PHx by RT-qPCR. Data are presented as mean ± s.e.m. (two-tailed Student's *t*-test. *n* = 5 mice per group). **c** Representative images of BrdU-positive cells of liver sections from *Zhx2*-WT and *Zhx2*-KO^hep mice at indicated time points after 2/3 PHx are displayed on the left panel. The quantitative data are presented on the right panel. Scale bar: 50 μm. Data are represented as mean ± s.e.m. (two-tailed Student's *t*-test. *n* = 5 mice per group). **d** The mRNA levels of proliferation-related genes, such as *CyclinD1, CyclinA2,*

*CyclinB1,* and *Cyclin E1,* were detected in the livers of *Zhx2*-WT and *Zhx2*-KO^hep mice after CCl4 injection, respectively. Data are presented as mean ± s.e.m. (two-tailed Student's *t*-test. *n* = 6 mice per group). **e** The plasma AST and ALT levels were determined in *Zhx2*-WT and *Zhx2*-KO^hep mice at indicated time points after CCl4 injection. Data are presented as mean ± s.e.m. (two-tailed Student's *t*-test. *n* = 4 mice per group). **f** Representative images of H&E staining for liver sections from *Zhx2*-WT and *Zhx2*-KO^hep mice after CCl4 injection are presented. Scale bar: 100 μm. The quantification data are displayed on the right panel. Data are presented as mean ± s.e.m. (two-tailed Student's *t*-test. *n* = 4 mice per group). **g** Representative images of TUNEL staining for liver sections from *Zhx2*-WT and *Zhx2*-KO^hep mice after CCl4 injection are displayed on the left panel, respectively. Scale bar: 50 μm. The quantification data are displayed on the right panel, respectively. Data are presented as mean ± s.e.m. (two-tailed Student's *t*-test. *n* = 6 mice per group).

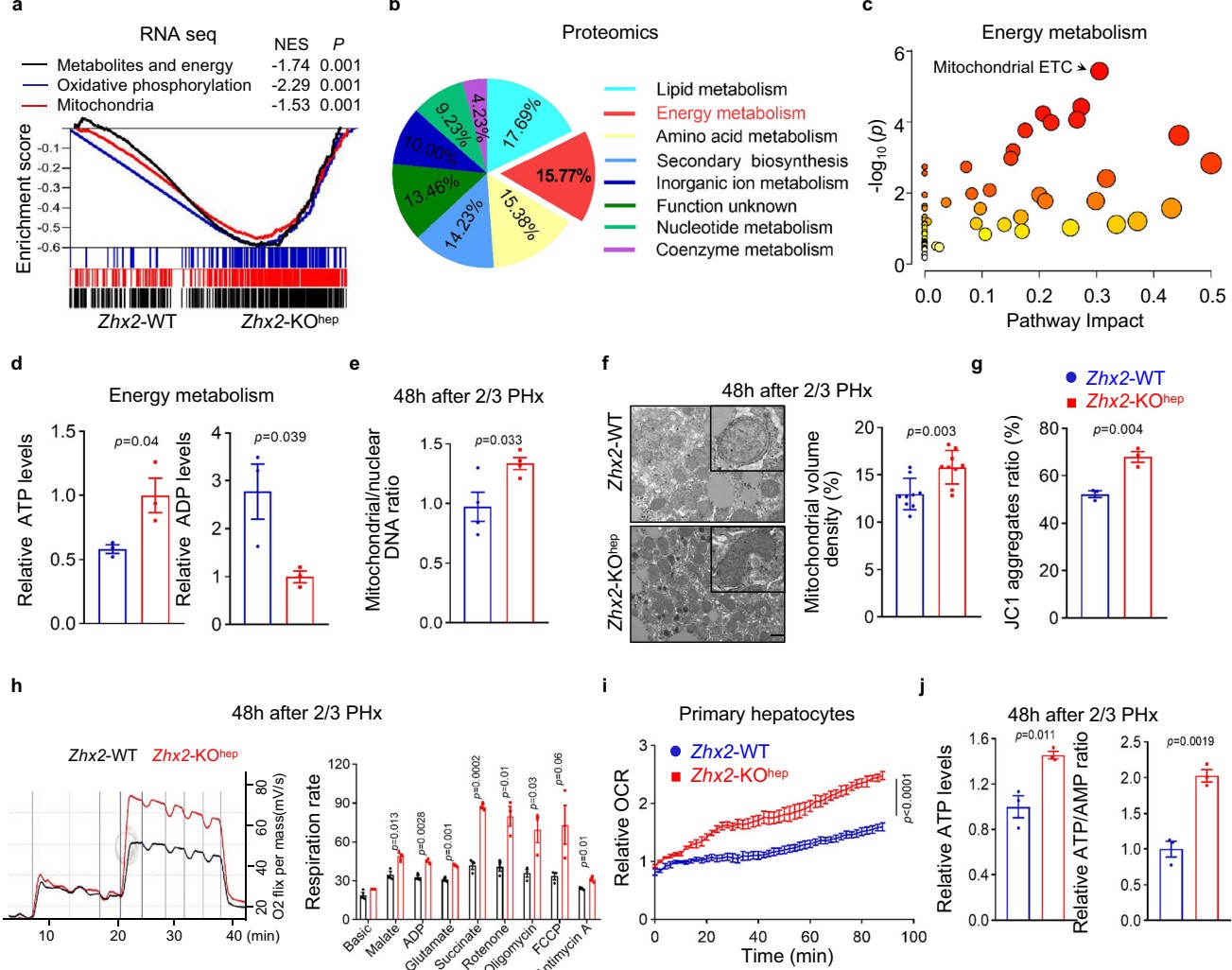

**Fig. 3 | Enhanced OXPHOS activity in hepatocytes with *Zhx2* deficiency during damaged liver repair. a–c** RNA-seq (**a**), quantitative proteomic analysis (**b**) and energy metabolomics (**c**) were performed with liver tissues from *Zhx2*-KO[hep] and *Zhx2*-WT mice at 48 h after 2/3 PHx. **a** GSEA was used to analyze the genes with differential mRNA levels. Data were analyzed using Kolmogorov–Smirnov test. **b** Percentage of per metabolic category among differential metabolic proteins was shown on the pie-chart. **c** Pathway analysis with MetaboAnalyst (www.metaboanalyst.ca/). Data were analyzed using Kolmogorov–Smirnov test (**a**), hypergeometric test (**c**). **d** The metabolites of ATP and ADP were determined by energy metabolic analysis in *Zhx2*-KO[hep] and *Zhx2*-WT mice liver samples. Data are presented as mean ± s.e.m. (two-tailed Student's *t*-test. *n* = 3 mice per group). **e** The copy number of mtDNA was determined in hepatocytes of *Zhx2*-KO[hep] and *Zhx2*-WT mice at 48 h after 2/3 PHx by qPCR. Data are presented as mean ± s.e.m. (two-tailed Student's *t*-test. *n* = 4 mice per group). **f** Ultrastructure of mitochondria in

hepatocytes from *Zhx2*-KO[hep] and *Zhx2*-WT mice were analyzed at 48 h after 2/3 PHx by using transmission electron microscopy (TEM). The representative images are shown on the left panel and one mitochondrial was selected to zoom in. The quantitative data of mitochondrial volume density is shown on the right panel. Scale bar: 1 μm. Representative data are presented as mean ± sd (two-tailed Student's *t*-test. *n* = 9 cells). **g–i** Mitochondrial functional status, including mitochondrial membrane potential (JC-1 aggregates) (**g**), extracellular O2 consumption (**h**), and Oxygen consumption (**i**) in hepatocytes from *Zhx2*-KO[hep] and *Zhx2*-WT mice 48 h after 2/3 PHx were accessed by the assay kits according to the manufacture protocols. Data are presented as mean ± s.e.m. (**g** and **h**: two-tailed Student's *t*-test. *n* = 3 mice; i: two-way ANOVA with Bonferroni's test. *n* = 4 mice). **j** ATP levels and ATP/AMP ratio were accessed in hepatocytes from *Zhx2*-KO[hep] and *Zhx2*-WT mice by the assay kits according to the manufacture protocols. Data are presented as mean ± s.e.m. (two-tailed Student's *t*-test. *n* = 3 mice).

change ≥ 1.2) between *Zhx2*-KO[hep] and *Zhx2*-WT mice at 48 h after 2/3 PHx, and KEGG analyses revealed metabolic pathway as one of top ten altered pathways in *Zhx2*-KO[hep] mice (Supplementary Fig. 3d). Further analysis of differentially expressed proteins in metabolic pathways showed that about 15.8% proteins were involved in mitochondria-related metabolic pathways, including energy metabolism, lipid and amino acid metabolism (Fig. 3b). Subsequently, energy metabolomics showed that the metabolites in mitochondrial ETC were enriched in the liver of *Zhx2*-KO[hep] mice (Fig. 3c), and the levels of ATP were significantly higher while the levels of ADP were lower in the liver of *Zhx2*-KO[hep] mice than *Zhx2*-WT mice at 48 h after 2/3 PHx (Fig. 3d).

Collectively, multi-omics data suggest that ZHX2 regulates mitochondrial OXPHOS during liver repair after acute injury.

To verify multi-omics findings, mitochondrial functional assays were performed in primary hepatocytes isolated from mice at 48 h after 2/3 PHx. As shown in Fig. 3e, mtDNA copy number was increased in primary hepatocytes of *Zhx2*-KO[hep] mice at 48 h after 2/3 PHx. Consistently, electron microscopy detected higher mitochondrial volume density in hepatocytes from *Zhx2*-KO[hep] mice (Fig. 3f). Subsequently, *Zhx2* knockout largely increased mitochondrial membrane potential (JC-1 aggregates), oxygen consumption and mitochondrial OXPHOS in primary hepatocytes isolated from mice at 48 h after 2/3 PHx

PHx (Fig. 3g–i). Similarly, the ATP levels, ATP/AMP and ATP/ADP ratios were higher while the AMP levels were lower in *Zhx2*-KO^hep hepatocytes than those of *Zhx2*-WT hepatocytes (Fig. 3j and Supplementary Fig. 3e). Altogether, loss of *Zhx2* promotes mitochondrial function during the repair of injured livers.

### *Zhx2* inhibits liver recovery by reducing OXPHOS

To further verify that *Zhx2* knockout facilitates repair of injured liver by regulating mitochondrial OXPHOS, metformin, a widely accepted

inhibitor that indirectly targets mitochondrial complex I and inhibits OXPHOS and ATP production[33,34], was used to treat Huh7 cells and mice. As shown in Fig. 4a, b, ZHX2 silence led to enhancement of mitochondrial membrane potential and AMP to ATP conversion in Huh7 cells, and these enhancements were abolished by metformin treatment. Importantly, liver regeneration and ATP production in metformin pretreated mice were assessed by following 2/3 PHx (Fig. 4c). As expected, metformin administration abolished the enhanced liver regeneration in *Zhx2*-KO^hep mice, as demonstrated by

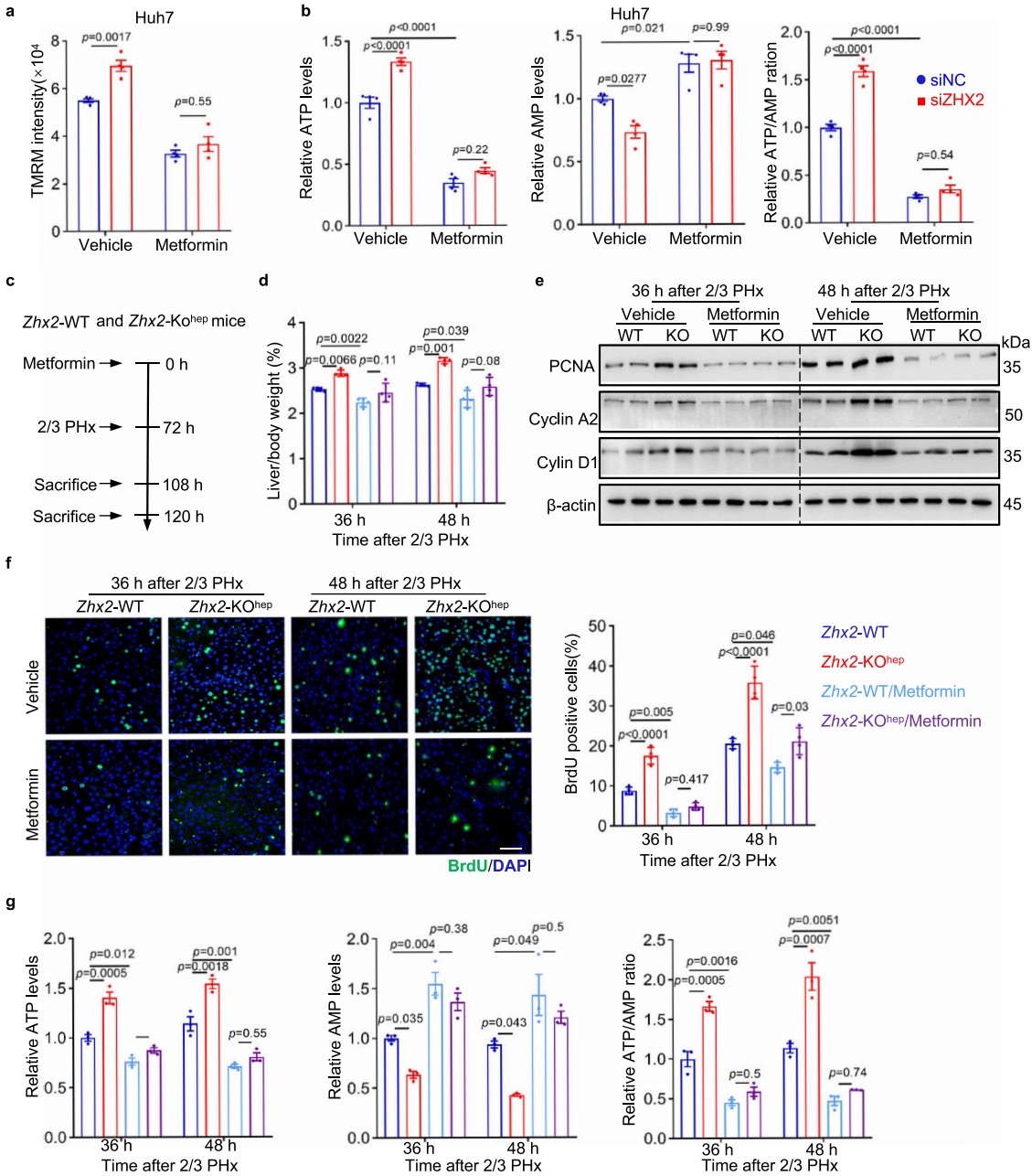

**Fig. 4 | Metformin treatment inhibits OXPHOS activity and dampens augmented liver regeneration in *Zhx2*-KO^hep mice. a**, **b** Huh7 cells were transfected with siNC and siZHX2, following with metformin or vehicle treatment. TMRM intensity (**a**), ATP and AMP levels, and ATP/AMP ratio (**b**) are shown. Representative data are presented as mean ± sd (two-way ANOVA with Tukey's test. *n* = 4; n biologically independent samples). **c** Diagram for metformin administration in *Zhx2*-KO^hep and *Zhx2*-WT mice with 2/3 PHx. **d** Liver/body weight ratios of *Zhx2*-WT and *Zhx2*-KO^hep mice with or without metformin administration were measured at 36 and 48 h after 2/3 PHx. Data are presented as mean ± s.e.m. (one-way ANOVA with Tukey's test. *n* = 4 mice per group). **e** Expression of PCNA, Cyclin A2 and Cyclin D1 in

*Zhx2*-WT and *Zhx2*-KO^hep livers with or without metformin treatment were obtained at 36 and 48 h after 2/3 PHx by western blot. **f** BrdU-positive cells in livers from metformin and vehicle-administrated *Zhx2*-KO^hep and *Zhx2*-WT mice were determined by immunofluorescence staining. Reprehensive images are shown on the left panels and the quantitative data are shown on the right panels. Scale bar: 50 μm. Data are presented as mean ± s.e.m. (one-way ANOVA with Tukey's test. *n* = 4 mice per group). **g** Levels of ATP and AMP, and ratio of ATP/AMP were determined in livers from *Zhx2*-KO^hep and *Zhx2*-WT at 36 h and 48 h after 2/3 PHx with/without metformin administration. Data are presented as mean ± s.e.m. (one-way ANOVA with Tukey's test. *n* = 3 mice per group).

the abrogation of increased liver/body weight ratio (Fig. 4d). Consistently, metformin treatment eliminated the increased levels of PCNA, Cyclin A2, Cyclin D1 proteins and increased numbers of BrdU-positive cells in *Zhx2*-KO[hep] mice (Fig. 4e, f). More importantly, metformin not only decreased ATP levels, but also eliminated the difference in AMP and ATP levels and ATP/AMP ratios in *Zhx2*-KO[hep] and *Zhx2*-WT mice (Fig. 4g), suggesting that metformin dampens *Zhx2* deficiency-accelerated liver repair by inhibiting ATP production.

To further exclude the non-specific effect of metformin and confirm the role of ZHX2-mediated OXPHOS inhibition in injured liver repair, *Zhx2*-KO[hep] and *Zhx2*-WT mice were pre-exposed to carbonyl cyanide-p-trifluoromethoxyphenylhydrazone (FCCP), an uncoupler of mitochondrial OXPHOS[33,35], following with 2/3 PHx (Supplementary Fig. 4a). Consistent with metformin treatment, FCCP treatment eliminated the differences in liver/body weight ratios, PCNA, Cyclin A2, Cyclin D1 expression, BrdU-positive cells, and ATP levels and ATP/AMP ratios in *Zhx2*-KO[hep] and *Zhx2*-WT mice after 2/3 PHx (Supplementary Fig. 4b–e). Taken together, these data demonstrate that ZHX2 inhibits liver repair by reducing OXPHOS.

### ZHX2 binds to the promoters of a subset of ETC genes for inhibition

To investigate the mechanisms by which *Zhx2* deficiency increases mitochondrial *OXPHOS*, we further analyzed the transcriptomic and proteomic data. As shown in Fig. 5a, *Zhx2*-KO[hep] increased hepatic expression of genes encoding mitochondrial complex I-V. Similarly, the heatmap of proteomic data showed the increased levels of ETC proteins in *Zhx2*-KO[hep] mice (Fig. 5b). In addition, mitochondria-located proteins also increase in *Zhx2*-KO[hep] mice liver (Supplementary Fig. 5a). RT-qPCR and western blot further confirmed the enhanced levels of ETC genes in liver tissues from *Zhx2*-KO[hep] mice 48 h after 2/3 PHx (Supplementary Fig. 5b, c). Furthermore, ZHX2 overexpression decreased the expression of ETC genes in Huh7 cells (Supplementary Fig. 5d), while knockdown down of ZHX2 increased the expression of ETC genes in human hepatic organoids (Fig. 5c). Above all, these data imply that ZHX2 regulates expression of ETC genes.

Since ZHX2 has been identified as a transcriptional factor, we performed ChIP sequencing (ChIP-seq) with cell lysate of ZHX2-HA overexpressing Huh7 cells (Supplementary Fig. 5e). KEGG analysis showed that ZHX2-bound genes were enriched in OXPHOS pathway (Fig. 5d), which was in agreement with RNA-seq data showing enrichment of mitochondrial OXPHOS genes in *Zhx2*-KO[hep] mice livers (Fig. 3a). To identify the direct target of ZHX2, we performed a cluster analysis of sharing genes in OXPHOS pathway obtained from RNA-seq data and ChIP-seq data. As shown in Fig. 5e, six ETC genes were overlapped in both sequencing data, indicating that these genes are the potential targets of ZHX2. Interestingly, ChIP-seq data showed that these ETC genes contain a common ZHX2-binding motif (*AGGCTGAG*) in their 5′ UTR. And, this potential ZHX2-binding motif also presented at the promoters of previously reported ZHX2 targeted genes (Supplementary Fig. 5f). In accordance, ChIP assays performed with anti-HA confirmed the occupancy of ZHX2 in the promoter regions of four ETC genes in Huh7 cells (Fig. 5f). Co-transfection and dual luciferase assays showed that ZHX2 overexpression in Huh7 cells markedly inhibited the promoter activities of *NDUFB9*, *SHDA*, *COX7C* and *UQCRC1*, which contain the ZHX2-binding motif (Supplementary Fig. 5g). Furthermore, when three repeated putative ZHX2-binding motifs were cloned into a luciferase reporter vector, results of dual luciferase assays demonstrated that ZHX2 overexpression decreased the luciferase activity in a dose-dependent manner (Fig. 5g). The interaction of ZHX2 with this motif was further illustrated by EMSA and pull-down assays using biotin-labeled ZHX2-binding motif as a probe. As shown in Fig. 5h, an increased gel shift was observed in ZHX2-overexpressing Huh7 cells (lane 3) as referred to that of control (lane 2). The shift was

weakened by non-labeled probes as specific competitor in a dose-dependent way (lanes 4 and 5). Further pull-down assay confirmed the interaction of ZHX2 with the biotin-labeled motif (Supplementary Fig. 5h). Hence, these observations support that ZHX2 binds to the promoter region of these ETC genes for repression.

### ZHX2 represses PGC-1α through FBXW7

Mitochondrial OXPHOS is controlled by nuclear- and mitochondrial-encoded proteins[36]. Since only a small subset of ETC genes contain ZHX2-binding motif, we therefore asked whether ZHX2 regulates mitochondrial OXPHOS activity through other intermediates. To address this, we screened the well-defined mitochondrial regulators, including PGC-1α, nuclear respiratory factor 1/2 (NRF1/2), and TFAM in RNA-seq data obtained from *Zhx2*-KO[hep] and *Zhx2*-WT livers after 2/3 PHx. Gene co-expression network analysis showed these factors in the central node of the ETC gene network (Supplementary Fig. 6a), in line with the critical roles of PGC-1α, NRF1/2 and TFAM in regulating mitochondrial functions[37–39]. Although ZHX2 had no significant effects on mRNA levels of *NRF1*, *TFAM* and *PGC-1α* (Supplementary Fig. 6b), ZHX2 overexpression largely decreased PGC-1α protein levels in Huh7 cells (Supplementary Fig. 6c). On the contrary, knockdown of endogenous ZHX2 increased PGC-1α protein levels in Huh7 cells (Supplementary Fig. 6d). Similarly, overexpression of ZHX2 decreased while knockdown of ZHX2 enhanced PGC-1α protein levels in HepG2 cells (Supplementary Fig. 6d). Importantly, liver-specific knockout of *Zhx2* enhanced PGC-1α protein levels in murine liver tissues (Fig. 6a). Similarly, ZHX2 knockdown increased PGC-1α protein levels in human liver organoids (Fig. 6b). All the data imply that ZHX2 regulates PGC-1α at the protein level.

To determine how ZHX2 reduces PGC-1α protein levels, stability of PGC-1α was monitored in Huh7 cells. ZHX2 overexpression decreased the half-life of PGC-1α (Fig. 6c). Treatment with the proteasome inhibitor MG132 but not the lysosome inhibitor CQ blocked PGC-1α reduction caused by ZHX2 overexpression in Huh7 cells (Fig. 6d), indicating that ZHX2 promotes proteasome-dependent degradation of PGC-1α. Consistently, results of in vitro ubiquitination assays showed that ZHX2 overexpression increased ubiquitination of PGC-1α in Huh7 cells in a ZHX2 level-dependent manner (Fig. 6e). Furthermore, GSEA analyses of RNA-Seq data revealed the enrichment of protein degradation genes sets in *Zhx2*-KO[hep] mice (Supplementary Fig. 6e). Heatmap showed the top E3 genes that were decreased in *Zhx2*-KO[hep] mice at 48 h after 2/3 PHx (Fig. 6f). Among them, FBXW7 is a reported E3 ligase targeting PGC-1α[40]. First, ZHX2-mediated down-regulation of FBXW7 was validated at mRNA level in mice livers (Fig. 6g). Consistently, ZHX2 overexpression increased FBXW7 and ZHX2 knockout decreased FBXW7 at mRNA and protein levels in Huh7 and HepG2 cells (Supplementary Fig. 6f, g). Moreover, ZHX2 not only suppressed FBXW7 promoter activity but also occupied the FBXW7 promoter region (Supplementary Fig. 6h–j). Importantly, in vitro ubiquitination assay demonstrated that FBXW7 knockdown abolished the increased ubiquitination of PGC-1α caused by ZHX2 overexpression (Fig. 6h). These findings demonstrate that ZHX2 promotes PGC-1α degradation via FBXW7.

### *Zhx2* suppresses OXPHOS and liver recovery by reducing PGC-1α

To determine whether ZHX2 inhibits mitochondrial OXPHOS via PGC-1α, a series of mitochondrial functional assays were performed in Huh7 cells with PGC-1α interventions. The efficiency of ZHX2 and PGC-1α overexpression and/or knockdown were verified by western blot, respectively (Supplementary Fig. 7a). As shown in Fig. 7a, b and Supplementary Fig. 7b, ZHX2 overexpression led to reduced mitochondrial mass and activity in Huh7 cells, and these reductions were reversed by PGC-1α overexpression, as evidenced by disrupted decreases in mtDNA content, mitochondrial mass, and mitochondrial membrane potential. Consistently, both seahorse assay (Fig. 7c) and

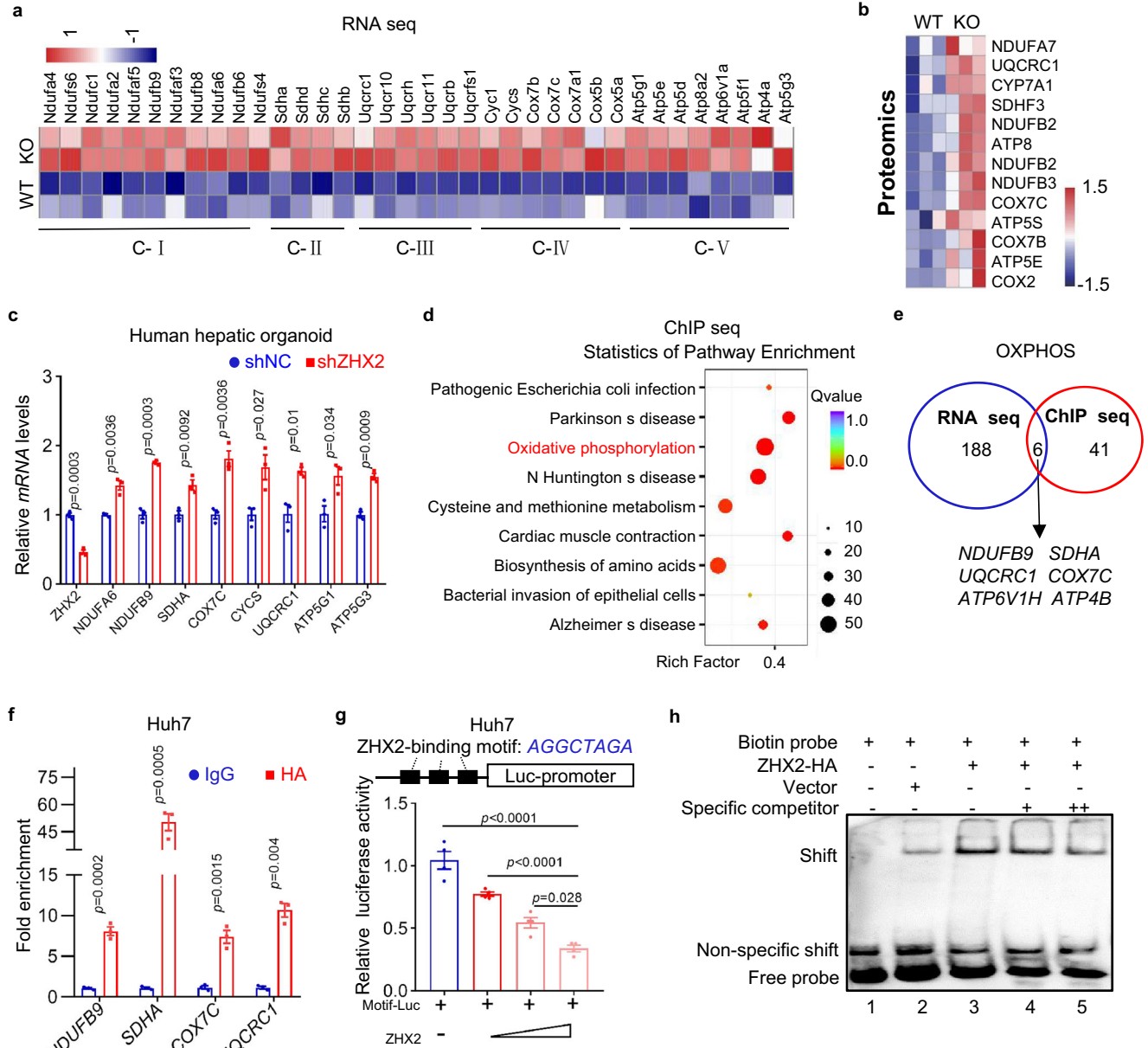

**Fig. 5 | ZHX2 transcriptionally represses ETC gene expression by binding to promoter regions. a**, **b** Heatmap from RNA-seq data and proteomics data of *Zhx2*-KO[hep] and *Zhx2*-WT mice livers at 48 h after 2/3 PHx. **a** Heatmap based on mRNA levels of mitochondrial complexes genes. **b** Heatmap showed proteins levels related to energy metabolism. Red and blue represent the increase and decrease of gene expression levels, respectively. **c** Relative mRNA levels of ETC genes were measured in human hepatic organoid cells transfected with LV-shZHX2 by RT-qPCR. Data are presented as mean ± sd (two-tailed Student's *t*-test. *n* = 3 biologically independent samples). **d** ChIP-seq was performed with HA antibody using cell lysate from ZHX2-HA overexpressing Huh7 cells. KEGG analysis showed top enriched gene sets based on ChIP-seq data. **e** Venn diagram showed genes in OXPHOS

pathway obtained from RNA-seq and ChIP-seq. **f** ChIP assay was performed with anti-HA antibody, IgG as control, using Huh7 cells transfected with ZHX2-HA. ZHX2 occupied at indicated genes were quantified by qPCR. Data are presented as mean ± sd (two-tailed Student's *t*-test. *n* = 3 biologically independent samples). **g** The luciferase reporter vector containing 3 putative ZHX2-binding motifs was co-transfected with a gradient dose of ZHX2-HA vectors in Huh7 cells. The reporter promoter activities were displayed. Data are presented as mean ± sd. (One-way ANOVA with Tukey's test. *n* = 4 biologically replicates). **h** EMSA were performed with nuclear extracts from Huh7 cells transfected with ZHX2-HA and control vectors. Biotin-labeled ZHX2-binding motif as probe and non-labeled motif as specific competitors.

Oxygraph-2k test (Supplementary Fig. 7c) demonstrated that ZHX2 reduced respiratory capacity of Huh7 cells, while PGC-1α co-transfection disrupted those differences. Reciprocally, ZHX2 knockdown increased the respiratory capacity of Huh7 cells, and this enhancement was greatly blocked by PGC-1α silence (Supplementary Fig. 7d). Also, ZHX2 knockdown elevated the mRNA levels of OXPHOS genes (*NDUFA6, CYCS, ATP5G1* and *ATP5G3*), which were abolished by PGC-1α silencing (Fig. 7d). Collectively, all above results reveal that ZHX2 regulates mitochondrial OXPHOS via PGC-1α.

To determine the role of PGC-1α in ZHX2-inhibited liver repair after acute damage, *Zhx2*-WT and *Zhx2*-KO[hep] mice were pretreated with PGC-1α inhibitor (SR18292), and were then subjected to 2/3 PHx (Fig. 7e). As shown in Fig. 7f, SR18292 administration almost completely abolished the difference of liver/body weight ratio between *Zhx2*-WT and *Zhx2*-KO[hep] mice. Consistently, SR18292 eliminated the *Zhx2* deficiency-induced increases in PCNA, Cyclin A2, Cyclin D1 proteins and the number of BrdU-positive cells in mice at 36 and 48 h after 2/3 PHx (Fig. 7g, h). In line with the evidence showing that ZHX2

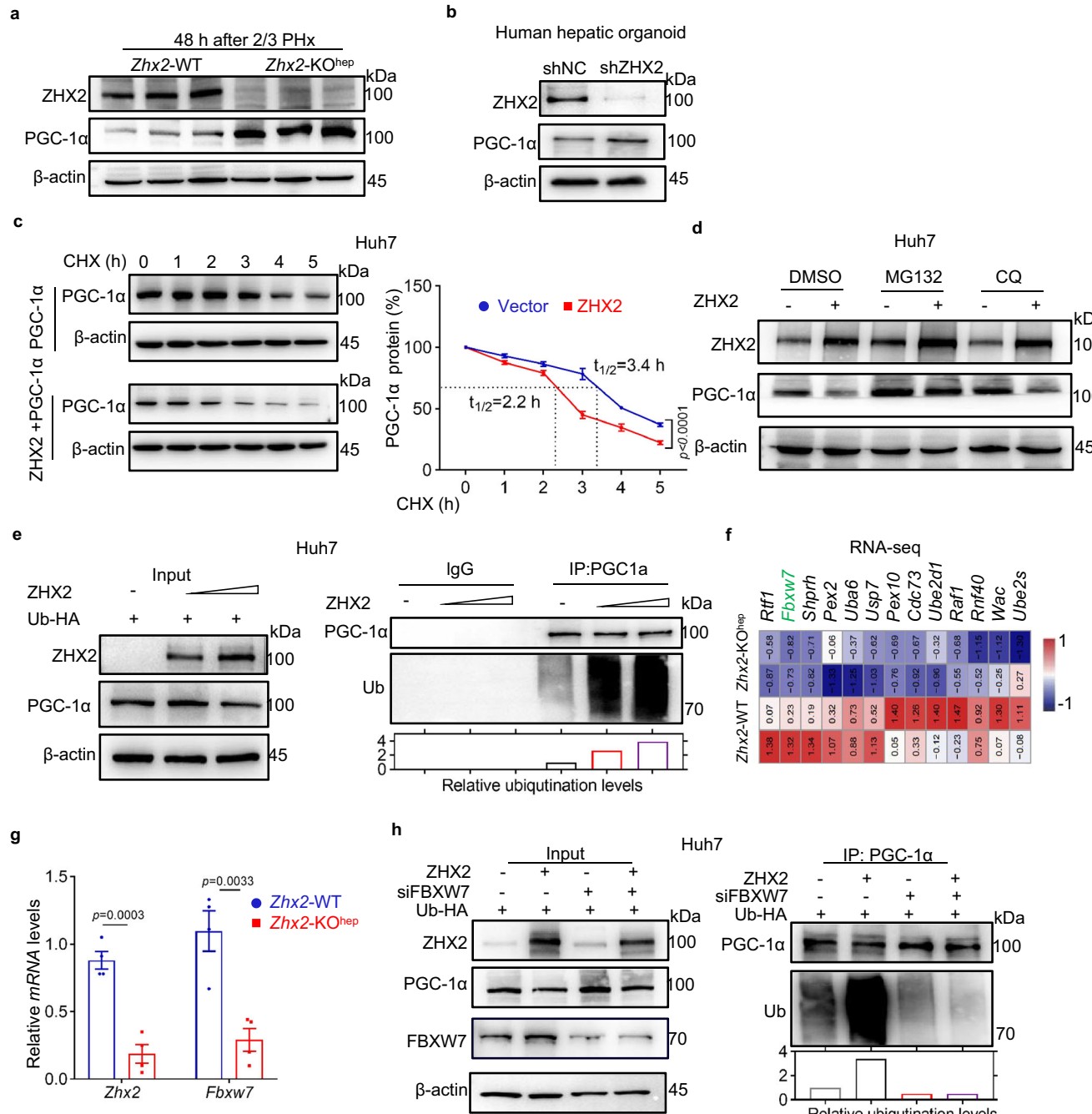

**Fig. 6 | ZHX2 promotes PGC-1α ubiquitination and degradation via FBXW7.**
**a**, **b** The expression of PGC-1α was detected in liver tissue from *Zhx2*-WT and *Zhx2*-KO[hep] mice at 48 h post 2/3 PHx (**a**) and human hepatic organoid with ZHX2 knockdown (**b**) by western blot. These experiments have been repeated for three times with similar results. **c** Cycloheximide (CHX, 500 μg/mL) chase assays were performed in Huh7 cells with or without ZHX2 overexpression for indicated time points. Representative immunoblots of PGC-1α are shown on the left panel. Quantification of relative protein abundance was displayed on the right panel. Data are presented as mean ± sd (two-way ANOVA with Bonferroni's test. *n* = 3 biologically independent samples). **d** Huh7 cells were transfected with or without ZHX2, following with MG132 or CQ treatment. Expression of PGC-1α was determined by

western blot. These experiments have been repeated for three times with similar results. **e** Ubiquitination of PGC-1α was measured in Huh7 cells co-transfected with HA-tagged ubiquitin and ZHX2 by western blot. These experiments have been repeated for three times with similar results. **f** Heatmap showed top 13 ubiquitination-related genes that decreased in *Zhx2*-KO[hep] mice at 48 h after 2/3 PHx. **g** Relative mRNA levels of *Zhx2* and *Fbxw7* were assessed in liver tissues of *Zhx2*-WT and *Zhx2*-KO[hep] mice at 48 h after 2/3 PHx by RT-qPCR. Data are presented as mean ± s.e.m. (two-tailed Student's *t*-test. *n* = 4 mice per group). **h** Ubiquitination of PGC-1α was evaluated by western blot in Huh7 cells transfected with HA-tagged ubiquitin, ZHX2 and siFBXW7 as indicated, respectively. These experiments have been repeated for three times with similar results.

suppressed OXPHOS during liver recovery, SR18292 treatment ameliorated the differences in ATP and AMP levels, and ATP/AMP ratios in *Zhx2*-KO[hep] and *Zhx2*-WT mice (Fig. 7i and Supplementary Fig. 7e). Above data *Zhx2*-KO accelerates damaged liver repair in PGC-1α-dependent manner.

## ZHX2 correlates with mitochondrial mass in DILI

To investigate whether ZHX2 is associated with susceptibility to liver toxicity and mitochondrial dysfunction in human, expression of ZHX2 and TOM20, a widely used mitochondrial mass marker, were examined by multiplexed immunofluorescence staining in liver biopsies from

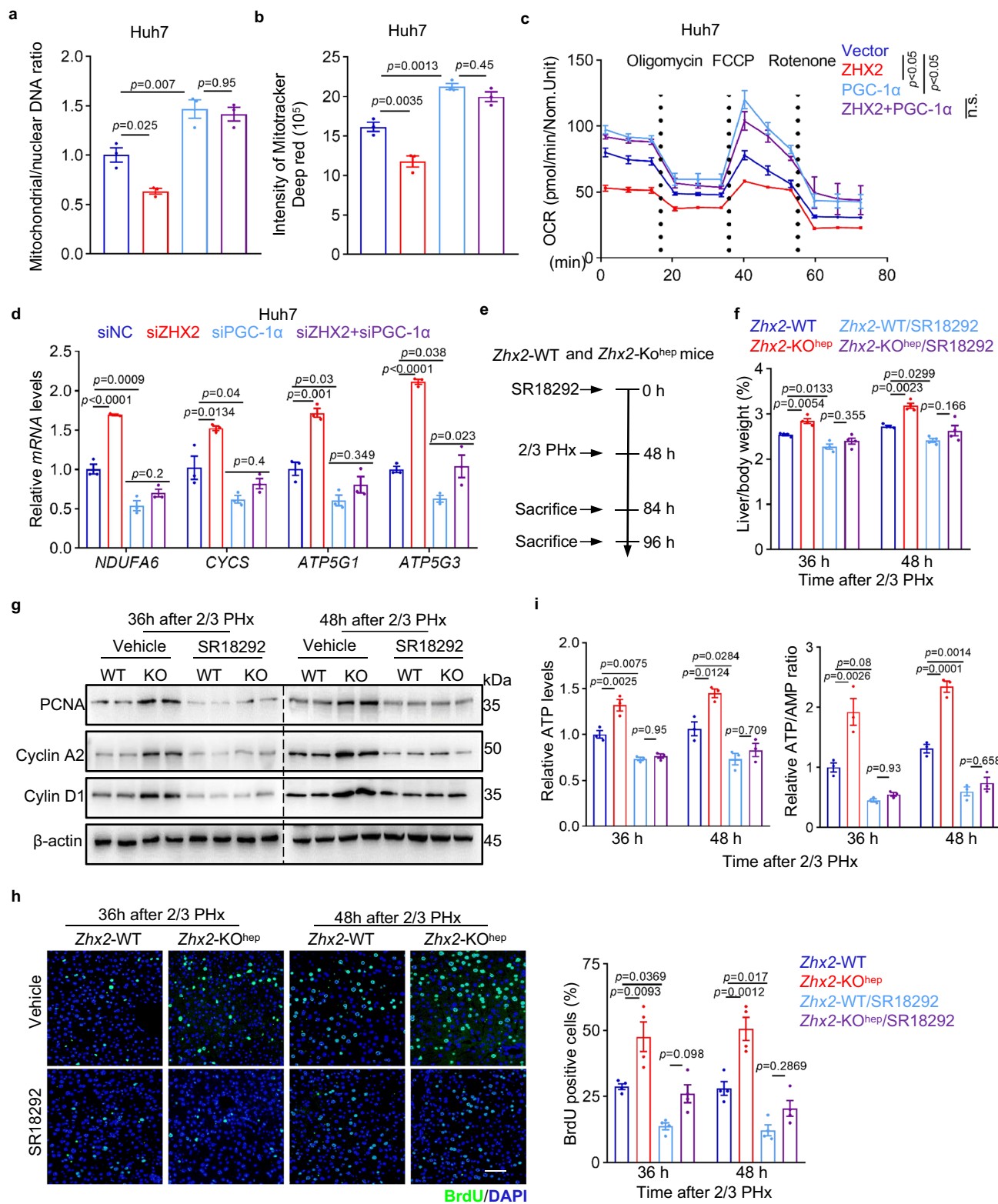

DILI patients, normal non-tumor sections from patients with hepatic hemangioma as normal control (Fig. 8a and Supplementary Table 1). In line with studies reporting drug-induced mitochondrial dysfunction in DILI[4,41], we detected decreased TOM20 in liver tissues from DILI patients (Fig. 8a–c). Notably, ZHX2 protein levels (fluorescence intensity) were higher in liver tissues from DILI patients than that of normal tissues, although ZHX2-positive cells were not significantly different in different groups (Fig. 8a, b). Furthermore, DILI patients

were classified into two groups based on their pathological characters[4]. Interestingly, ZHX2 levels were higher and TOM20 levels were lower in liver tissues from patients with severe DILI than those with mild DILI (Fig. 8a–c). It is worth noting that ZHX2 expression was negatively associated with TOM20 intensity in liver biopsies from DILI patients (Fig. 8d), indicating that ZHX2 was negatively associated with mitochondrial mass. Similar results were obtained by staining mitochondria with another marker COX IV (Supplementary Fig. 7f). These

**Fig. 7 | Loss of ZHX2 promotes mitochondrial OXPHOS and liver repair via PGC-1α. a, b** Mitochondrial mass was detected in Huh7 cells transfected with ZHX2 and/or PGC-1α by measuring mtDNA copy number (**a**) and intensity of Mito Tracker deep red (**b**). Data are presented as mean ± sd (two-way ANOVA with Tukey's test. $n = 3$ biologically independent samples). **c** Oxygen consumption rates (OCR) in ZHX2 and/or PGC-1α transfected Huh7 cells were evaluated by seahorse analyzer. Representative data are presented as mean ± sd (two-tailed Student's *t*-test. $n = 3$ biologically independent samples. n.s. indicates the difference is not significant). **d** The expression of ETC genes were analyzed in Huh7 cells transfected siZHX2 and/or siPGC-1α by RT-qPCR, respectively. Representative data are presented as mean ± sd (one-way ANOVA with Tukey's test. $n = 3$ biologically independent samples). **e–i** *Zhx2*-WT and *Zhx2*-KO[hep] mice were pretreated with PGC-1α inhibitor SR18292 or vehicle, following with 2/3 PHx. Mice livers were collected at 36 and 48 h after 2/3 PHx. Panel (**e**) shows the work flow. Panel (**f**) displays the liver/body weight ratios after 2/3 PHx ($n = 4$ mice per group, mean ± s.e.m.). Panel (**g**) presents protein levels of cell proliferated genes. Panel (**h**) shows BrdU-positive cells. Representative Images (left) and quantitative data (right) are shown. Scale bar: 50 μm. Data are presented as mean ± s.e.m. ($n = 4$ mice per group). Panel (**i**) shows the ATP levels and ATP/AMP ratios. Data are presented as mean ± s.e.m. (one-way ANOVA with Tukey's test. $n = 3$ mice per group).

data reveal a negative correlation between hepatic ZHX2 expression and drug-induced hepatic mitochondrial mass in human liver injury.

The elevated ZHX2 in livers with DILI suggests ZHX2 as a therapeutic target. To evaluate the potential application of silencing ZHX2 in the repair of acutely damaged liver, we mimicked DILI by exposing mice to CCl4. Mice were injected with pSilencer-sh*Zhx2* and control vectors before CCl4 exposure. Western blot confirmed the successful silence of hepatic *Zhx2* mediated by sh*Zhx2* injection (Fig. 8e). H&E results showed that the injured area of mice livers was obviously less in mice treated with sh*Zhx2* than that of control mice (Fig. 8f). Consistently, the numbers of Ki67-positive cells were higher and the numbers of TUNEL-positive cells were lower in mice liver with *Zhx2* knockdown (Fig. 8g, h). Furthermore, the recovery of liver function was accelerated, displaying as the levels of serum ALT, AST, ALP, TBL, TBA and GGT were all significantly lower in *Zhx2*-knockdown mice than that of control mice (Fig. 8i). Above all, ZHX2 silencing contributes to augmented repair of mice liver after CCl4-induced acute injury.

## Discussion

Acute liver injury, characterized by a rapid loss of functional hepatocytes, severely threatens human health. If not treated promptly, it can easily lead to liver failure, a lethal outcome that currently lacks effective treatment, within days or weeks[42,43]. Mitochondria provide energy for rapid proliferation of hepatocyte[3]. Therefore, identifying mitochondrial regulators and deep understanding their regulation on mitochondria may pave the way for the future development of therapeutic strategies in liver injury. Here, we found that *Zhx2* loss in hepatocytes led to enhanced mitochondrial OXPHOS via PGC-1α dependent and independent manners, which contributes to accelerated repair of acute injured liver. Importantly, ZHX2 expression is negatively associated with mitochondrial marker TOM20 and COX IV in liver tissues from DILI patients. Previous research found that ZHX2 carried out its transcriptional activity upon reaching certain threshold[44]. Therefore, strategies aiming to manipulate ZHX2 may be a helpful strategy to enhance injured liver repair.

ZHX2 was first identified as a ubiquitous transcription factor (TF) interacting with NF-YA[45], and later studies have demonstrated the important roles of ZHX2 in different biological processes, including development, metabolism, and cancer[24,46,47]. Interestingly, ZHX2 works as a tumor suppressor in liver cancer and thyroid cancer[24,48], but functions as an oncogene in clear cell renal cell carcinoma (ccRCC)[49], which strongly suggests that ZHX2 takes effects in a context-dependent manner. Here, we clarified ZHX2 as a mitochondrial negative regulator in acute injured liver. Using in vitro, ex-in vivo and in vivo models, our data demonstrated that overexpression of ZHX2 reduced mitochondrial mass, membrane potential, OXPHOS. Reciprocally, knockdown or knockout ZHX2 increased mitochondrial activity displaying as above maintained parameters. Consistently, our previous work showed loss of Zhx2 enhanced lipid accumulation to promote NAFLD progression and increased mitochondria-mediated lipid oxidation to provide ATP for rapid hepatocytes proliferation during NAFLD-HCC transition[27]. Recently, Zhao et al. reported that deletion of

*Zhx2* augments NASH progression by enhancing PTEN-mTOR-dependent lipogenesis[29]. Since mitochondrial alterations play an important role in fatty liver diseases[7], ZHX2-mediated OXPHOS regulation might also involve in NASH progression. However, mitochondrial oxidative capacity varies broadly across the spectrum of obesity and NAFLD[7], and the role of mitochondrial in NASH progression is controversial. For instance, improving mitochondrial fatty acid oxidation can prevent nonalcoholic steatohepatitis progression[50]. On the contrary, hepatocyte-specific deletion of AIF indicated that the loss of mitochondrial OXPHOS protected against diet-induced steatosis and NASH progression[10]. Therefore, mitochondrial dysfunction in humans is manifest as a variety of disorders with clinical outcomes largely dependent on the magnitude and tissue distribution of the impairment[51,52]. Therefore, the role of ZHX2-mediated OXPHOS regulation in metabolic liver diseases requires further studies. Nevertheless, ZHX2 is identified as a negative mitochondrial regulator, manipulation of which could be beneficial for therapy of acute liver injuries.

Being a ubiquitous transcription factor, ZHX2 transcriptionally represses expression of many genes, such as AFP, GPC3 and H19[22,25]. Importantly, ZHX2 had been reported to transcriptionally repress expression of Cyclin A and E[24], the well-known cell cycle regulators during liver regeneration. However, ZHX2-binding element on its target gene promoter is still unclear. Here, by integrated analyses of RNA-seq and ChIP-seq data, we identified a putative ZHX2-interacting consensus motif, which is in the 5'-UTR of some ETC genes. Both luciferase activity and ChIP-qPCR assays demonstrated that ZHX2 occupies on ETC gene promoters via this consensus motif, leading to transcriptional repression. Notably, EMSA and pull-down assays confirmed that ZHX2 binds to the biotin-labeled consensus motif. Previous reports showed that ZHX2 interacts with NF-YA or RelA/p65 to repress or promote gene expression[45,49]. To the best of our knowledge, the present work is identifying the direct binding motif of ZHX2. It is worth noting that the putative ZHX2-binding motif is also located in the previously reported ZHX2 target genes, such as *Cyclin E* and *KDM2A*, which are involved in liver cancer progression[24,53]. These findings could shed light on the research about the function of ZHX2 in genes transactional regulation.

Another mechanism by which ZHX2 represses OXPHOS is that ZHX2 reduces PGC-1α protein levels by enhancing FBXW7 transcriptional activation, thereby limiting mitochondrial biogenesis. PGC-1α is dynamically regulated at the mRNA and protein levels in response to various signaling pathways involved in cellular growth, differentiation, and energy metabolism[39,54]. Previous report showed that PGC-1α is rapidly and dramatically induced at transcriptional level after hepatectomy[55]. However, its post-transcriptional regulation after hepatectomy is still unclear. Here, using protein half-life and ubiquitination assays, we demonstrated that ZHX2 regulates PGC-1α at protein levels. Importantly, FBXW7, an E3 ligase of PGC-1α, was transcriptionally upregulated by ZHX2[40]. Extensive body of literature implicates that FBXW7 mediates ubiquitination and degradation of c-Myc[56,57], which participates in the "priming" of hepatocytes after partial hepatectomy[58]. In addition, ZHX2 promotes PTEN

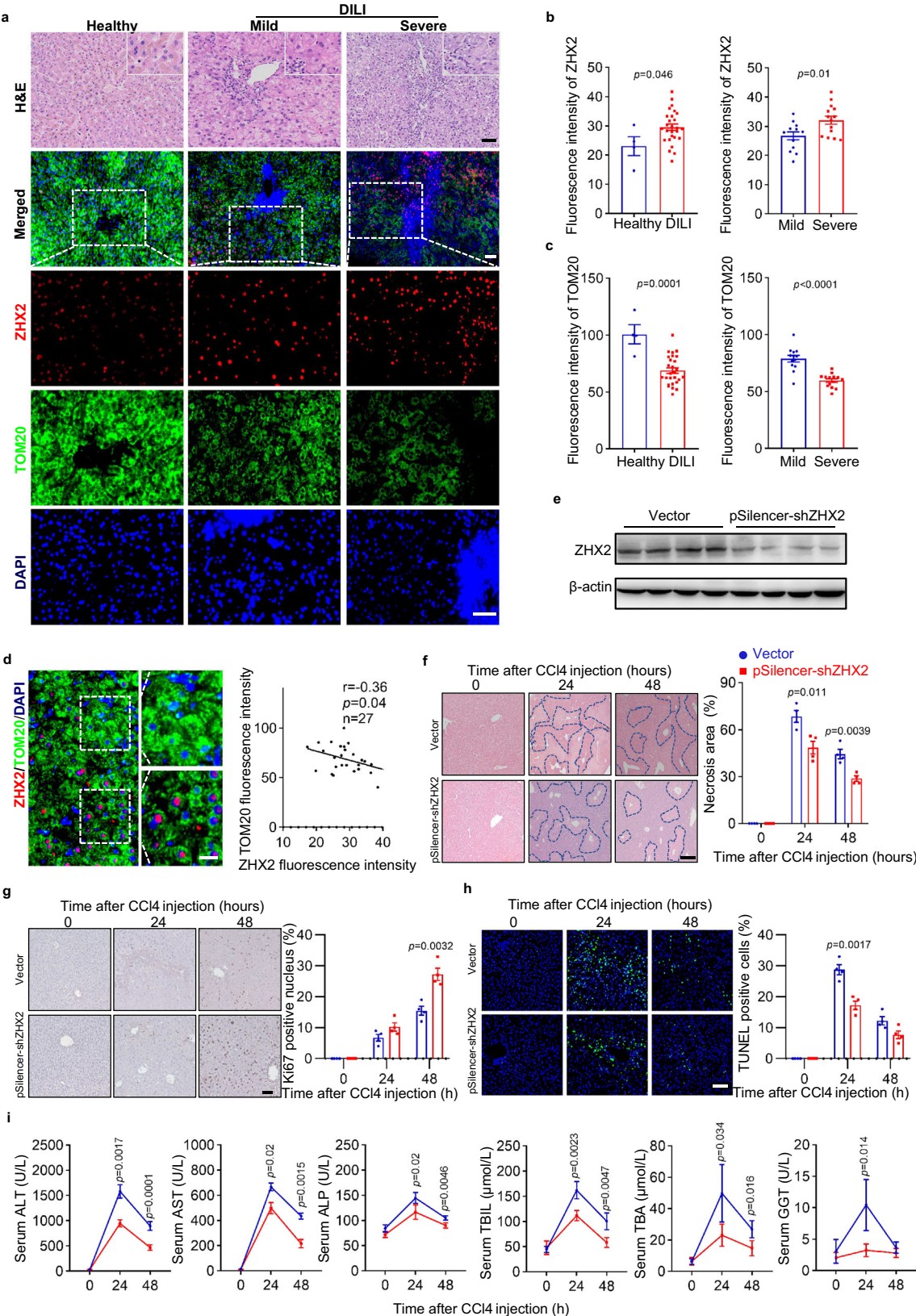

transactivation to inhibit mTOR signaling and suppresses LPL to block exogenous lipids uptake[29,46], which are also associated with mitochondrial activity. Thus, ZHX2 involves in the repair of acute liver injuries through complex pathways and signals. Nevertheless, based on our findings and previous reports, in *Zhx2* knockdown or knockout mice, PGC-1α-dependent mitochondrial OXPHOS is the most

significant changed pathways that contributes to accelerated liver recovery after acute injuries.

In conclusion, our work identified ZHX2 as a regulator of mitochondria function that contributes to repair of injured livers. On one hand, ZHX2 transcriptionally represses ETC gene expression by binding to consensus motif in their promoter. On the other hand, ZHX2

**Fig. 8 | ZHX2 expression is increased in human DILI and silencing ZHX2 accelerates liver recovery in mice with CCl4-induced injury. a–d** Human samples from normal non-tumor sections of patients with hepatic hemangioma (Healthy) (*n* = 4) and from patients with drug-induced liver injury (DILI) (*n* = 27) were used for examination. **a** Representative H&E images and IF staining of ZHX2 and TOM20 of DILI patients and healthy donors were displayed. Scale bar: 50 μm. **b, c** showed the quantified data. Data are presented as mean ± sd. Two-tailed Student's *t*-test. **d** Correlation analysis of fluorescence intensity of ZHX2 and TOM20 in DILI patients. Right, representative images. Left, quantitative data. Scale bar: 20 μm. Data are represented as mean ± s.e.m. Pearson's correlation coefficients (r) and *p* values (*p*) for two-sided correlation tests are shown. **e–h** C57BL/6 mice were injected with Vector or pSilencer-sh*Zhx2* via the tail vein hydrodynamic injection, respectively. Five days later, the mice were used to induce liver injury by CCl4 injection. **e** Expression of ZHX2 in mice liver with or without *Zhx2* knockdown was determined by western blot. Representative images of H&E staining (**f**), Ki67-positive cells (**g**) and TUNEL-positive cells (**h**) of liver sections at indicated time points after CCl4 injection are displayed at left panel. The quantitative data are presented on the right panel. **f** Scale bar: 100 μm. **g, h** Scale bar: 50 μm. Data are represented as mean ± s.e.m. (two-tailed Student's *t*-test. *n* = 4 mice per group). **i** The plasma ALT, AST, ALP, TNIL, TBA and GGT levels were determined at indicated time points after CCl4 injection. Data are presented as mean ± s.e.m. (two-tailed Student's *t*-test. *n* = 4 mice per group).

destabilizes PGC-1α protein to reduce mitochondrial biogenesis and OXPHOS. Thus, the present research has not only improved our understanding of ZHX2 biological activities, but also provides a potential strategy for treatment of acute liver injuries, such as DILI, and liver resection.

## Methods

### Human samples
A total of 27 samples from patients with drug-induce liver injury (DILI) were collected from Qilu Hospital of Shandong University (*n* = 12) and Beijing Ditan Hospital Capital Medical University (*n* = 15). Drugs associated with DILI are angiotensin-converting enzyme inhibitor (1/27, 3.7%), antidiabetic agents (2/27, 7.4%), anticonvulsant drugs (1/27, 3.7%), dietary supplements (3/27, 11.1%), Chinese herbal medicine (12/27, 44.4%), antibiotics (1/27, 3.7%), acetaminophen (2/27, 7.4%), antipsychotic agents (1/27, 3.7%), antihistamines (1/27, 3.7%). There are three patients whose medication information is not available, but liver histology provided information supporting the diagnosis of DILI. All the samples were separated into three groups: hepatocellular injury, cholestatic injury and mixed injury, according to types of liver damage. Adjacent non-tumoral tissues from patients with hepatic hemangioma in Qilu Hospital, Shandong University, were used as normal liver samples (*n* = 4). Diagnosis of DILI was determined by three pathologists from the above two hospitals. According to the grade of damaged severity, samples were grouped into mild, moderate, severe and fatal[59]. The detailed pathological information of these samples is described in Supplementary Table 1. Histological analysis and genes expression detection were presented in the following part. These studies were approved by the Research Ethics Committees of Qilu Hospital, Shandong University (KYLL-202209-033) and Beijing Ditan Hospital Capital Medical University (2021-041-01). All patients gave informed consent for all clinical investigations, according to the principles embodied in the Declaration of Helsinki.

### Mice
C57BL/6 mice (6–8 weeks of age) were purchased from GemPharmatech LLC. Albumin-Cre mice were obtained from Jackson Laboratory. *Zhx2* floxed mice were gifted by Prof. Brett T Spear from University of Kentucky, hepatocyte-specific *Zhx2* deficient mice (*Zhx2*-KO^hep) were generated by crossing Albumin-Cre and *Zhx2* floxed mice[60]. Their littermates without Albumin-Cre were defined as *Zhx2*-WT mice. All mice were maintained under specific pathogen-free conditions with a 12-h light, 12-h dark cycle and given free access to food (Xiaoshuyoutai, AIN-93M) and water at Shandong University Laboratory Animal Center. Experiments were carried out under the Shandong University Laboratory Animal Center's approval. Animal Ethics Number: ECSBMSSDU2018-1-031.

### Partial hepatectomy and CCl4 injection
For the partial hepatectomy model, 8–12-week-old mice were randomly anesthetized with 1% pentobarbital sodium. Then the left lateral and median hepatic lobes were ligated and removed. After closing the abdominal cavity, the sutured incision was sterilized with betadine, and the mice were placed on a warming pad for regeneration. At the indicated time points, the mice were sacrificed, and the mice livers were collected. The liver/body weight ratios were calculated. Then, harvested liver tissues were either fixed in buffered formalin or snap frozen in liquid nitrogen and stored at −80 °C until use.

For CCl4-induced mice liver injury, 8–12-week-old hepatic-specific *Zhx2* knockout and littermate control mice were treated with CCl4 (10% 10 μL/g). Then the mice liver tissues were collected at 24, 48 and 72 h after CCL4 injection for assessment as above.

To evaluate whether Zhx2 silencing is effective to promote recovery of drug-induced liver damage, 8–12-week-old C57BL/6 mice were injected with pSilencer-sh*Zhx2* (15 μg) or control vectors via the tail vein hydrodynamic injection, respectively. Five days later, the mice were treated with CCl4 (10% 10 μL/g). Then the mice were sacrificed at 24 and 48 h after CCl4 treatment. Liver tissues and blood serum were collected to examine recovery of liver injury and function.

### Quantitative real-time PCR (RT-qPCR)
Total RNA of cells, liver tissues and organoids were extracted using TRIzol reagent (TIANGEN Biotech, DP424) and reverse transcribed into cDNA with RevertAid First Strand cDNA Synthesis Kit (Thermo Fisher Scientific, K1622). qPCR was carried out using a BioRad C1000 Thermal Cycler CFX96 Real-Time System with ChamQ Universal SYBR qPCR Master Mix (Vazyme Biotech, Q711). Primer for RT-qPCR, see Supplementary Table 2.

### Western blot
The total proteins of liver tissue from *Zhx2*-WT and *Zhx2*-KO^hep mice and Huh7 cells were collected in cell lysis buffer (Beyotime, P0013). And protein extracts were quantified by BCA protein assay (Beyotime, P0009). Equal of protein were loaded in SDS-polyacrylamide electrophoresis gel, transferred to Immobilon-P Membranes (Millipore, IPVH00010) and incubated overnight at 4 °C with the following primary antibodies from Proteintech: anti-β-actin (66009-1-Ig, 1:5,000; RRID:AB_2687938), anti-ZHX2 (20136-1-AP, 1:4,000; RRID: AB_10666438), anti-PCNA (10205-2-ap, 1:4,000; RRID: AB_2160330), anti-Cyclin A2 (18202-1-AP,1:2,000; RRID: AB_10597084), anti-Cyclin D1 (60186-1-Ig,1:2,000; RRID: AB_10597084), anti-NRF1 (12482-1-AP,1:2,000; RRID: AB_2282876), anti-TFAM (22586-1-AP,1:2,000; RRID: AB_11182588), anti-PGC-1α (66369-1-Ig,1:2,000; RRID: AB_2828002), anti-NDUFB9 (15572-1-AP,1:1,000; RRID:AB_2267110), anti-COX7C (11411-2-AP,1:1,000; RRID:AB_2085713), anti-SDHA (14865-1-AP,1:2,000; RRID:AB_11182164), anti-UQCRC1 (21705-1-AP,1:2,000; RRID:AB_10734437). CST: anti-Cyclin B1 (4138T,1:1,000), anti-Cyclin E1 (20808s,1:1,000; RRID: AB_2783554). Abcam: anti-FBXW7 (ab109617,1:1,000; RRID: AB_2687519). The membranes were washed with PBST for 3 times and subsequently incubated with secondary HRP-conjugated anti-mouse (Proteintech, SA00001-1, 1:5,000) or anti-rabbit IgG secondary antibodies (Proteintech, SA00001-2, 1:5,000). The signal was detected by enhanced chemiluminescence (ECL)

reagent (Millipore, WBULS0500) using the Tanon Bio-Imaging Systems (Tanon, 4600).

## Cell lines, primary hepatocytes, human liver organoids

Huh7(SCSP-526), HepG2(SCSP-510) and HEK293T(SCSP-502) were purchased from Shanghai Institute of Cell Biology (Chinese Academy of Sciences, China) and cultured in DMEM (GIBCO, C11995500BT) supplemented with 10% FBS (BI, 10270-106) at 37 °C. Metformin (Sigma-Aldrich, D150959) and SR18292(MCE, HY-101491).

Primary hepatocytes were cultured in Williams' Medium E (Thermo Fisher Scientific, 22551089) supplemented with 5% FBS, 0.5% penicillin/streptomycin (Solarbio, p1400) and 15 mM HEPES (Thermo Fisher Scientific,15630080) at 37 °C in a 5% $CO_2$ incubator overnight before use in experimentation.

Human liver organoids were cultured in the medium: AdDMEM/F12 (Thermo Fisher Scientific, 12634028), 0.5% Penicillin-Streptomycin, 1% GlutaMAX (Thermo Fisher Scientific, 35050061), $10 \times 10^{-3}$ M HEPES, 1% B27 minus vitamin A, 15% R-spodin1-conditioned medium, $3 \times 10^{-6}$ M ChIR99021 (Sigma-Aldrich, SML1046), $10 \times 10^{-3}$ M nicotinamide (Sigma-Aldrich, N0636), 50 ng mL$^{-1}$ EGF (Peprotech,100-47), 20 ng mL$^{-1}$ TGF-α (Peprotech, 100-16A), 100 ng mL$^{-1}$ FGF7 (Peprotech, 100-19), 50 ng mL$^{-1}$ HGF (Peprotech, 100-39H), $1 \times 10^{-6}$ M dexamethasone (Sigma, D4902), 10 ng mL$^{-1}$ Oncostatin M (Sigma, O9635).

## Plasmids

Polymerase chain reaction (PCR)-amplified human PGC-1α was cloned into pcDNA3.0-Flag vector. ZHX2-expressing vectors ZHX2 and Tet-On-ZHX2 have been described previously[27]. The pSilencer-ZHX2 expression plasmid was constructed and maintained in our laboratory. Lentivirus expressing shRNA against ZHX2 and the control lentivirus bought from GenePharma were included for in vitro. Promoter regions of ETC genes were amplified using the specific primers and cloned to pGL3-Promoter vectors (Promega, E1761) to construct the luciferase report vectors, respectively.

## Mouse experiments

For the metformin-administered 2/3 PHx mice model, the *Zhx2*-KO$^{hep}$ and *Zhx2*-WT mice were intraperitoneally injected with metformin (400 mg/kg/day in PBS) and vehicle (PBS). Seventy-two hours later, these mice were operated with liver resection. Then, the mice were sacrificed to calculate liver/body weight ratios and collect liver tissues at 36 and 48 h after 2/3 PHx.

For the FCCP-administered 2/3 PHx mice model, the *Zhx2*-KO$^{hep}$ and *Zhx2*-WT mice were intraperitoneally injected with FCCP (2 mg/kg/day in DMSO) and vehicle (DMSO). Seventy-two hours later, these mice were operated with liver resection. Then, the mice were sacrificed to calculate liver/body weight ratios and collect liver tissues at 36 and 48 h after 2/3 PHx.

For PGC-1α activity inhibited 2/3 PHx mice model, the *Zhx2*-KO$^{hep}$ and *Zhx2*-WT mice were intraperitoneally injected with SR18292 (50 mg/kg/day in 2% DMSO in PBS) and vehicle (2% DMSO in PBS). Forty-eight hours later, 2/3 of these mice livers were surgically removed. After 36 and 48 h, the mice were sacrificed to calculate liver/body weight ratios and collect liver tissues.

## Histological analysis

Liver tissues were fixed with 4% paraformaldehyde for 24 h and embedded in paraffin. The 4-μm paraffin sections were deparaffinized in xylene and concentrations of ethanol (100−50%). The sections were stained with standard hematoxylin and eosin (H&E) procedure. A blinded observer was assigned to evaluate the histopathological liver injury of mice livers. Histopathology of samples from patients with DILI was determined by three pathologists from Qilu Hospital of Shandong University and Beijing Ditan Hospital Capital Medical University.

## TUNEL assay and Ki67 staining

Terminal deoxynucleotidyl transferase-mediated deoxyuridine triphosphate nick-end labeling (TUNEL) staining was performed on paraffin-embedded tissue sections using the In Situ Cell Death Detection Kit, Fluorescein (Roche, 11684795910). TUNEL-positive areas were quantified with ImageJ.

For anti-Ki67 staining, deparaffinized tissue sections were treated with sodium citrate antigen retrieval solution (Solarbio, C1032) in a microwave oven to retrieve antigen. Then, tissue sections were blocked with 5% BSA (Solarbio, A8020) in PBS for 30 min. Liver sections were incubated with polyclonal antibody against Ki67 (Abacm, ab15580, 1:200) in PBS with 1% BSA overnight at 4 °C followed by incubating for 1 h at room temperature with a biotinylated anti-rabbit IgG secondary antibody (Dako, K5007). At last, the positive signal was detected by using Diaminobenzidine (DAB) Histochemistry Kit (Dako, K5007). The images were captured and digitalized using an Olympus microscope attached to an Olympus digital camera.

## Tissue immunofluorescence and multiplexed immunofluorescence staining

The liver sections were deparaffinized by the same protocol as the Ki67 staining. Briefly, the sections were incubated with 0.25% Triton-X-100 in PBS for 10 min at room temperature followed by incubating in a blocking buffer (4% BSA, 2% serum in PBS) for 1 h at room temperature. The sections were then incubated with diluted anti-BrdU (Abacm, ab6326) in blocking buffer overnight at 4 °C. After washing in PBS, the slides were further incubated with fluorescein-conjugated secondary antibodies diluted in blocking buffer for 1 h at room temperature. Finally, the sections were mounted with ProLong Diamond Antifade Mountant with DAPI (Beyotime, C1002) to examine and capture images on a Leica microscopy.

Multiplexed immunofluorescence staining of samples from patients with DILI was performed using Opal Chemistry (AKOYA, NEL861001KT) with antibodies. In brief, after deparaffinization, slides were processed with microwave (4 min 100% power, 15−20 min 20% power) in antigen retrieval buffer, and blocked with antibody diluent for 10 min at room temperature. Slides were incubated with the primary antibody for 30−60 min, and subsequently incubated with HRP-conjugated secondary antibody for 10 min after removing the primary antibody and washing in TBST buffer. Thereafter, slides were incubated with Opal working buffer for 10 min at room temperature and then washed in TBST buffer. The above procedures were repeated for other antibodies, and antibodies were removed by microwave treatment (45 s 100% power, 15−20 min 20% power) before another round of staining was performed. Finally, we used DAPI to highlight all nuclei.

## Super-resolution structured illumination microscopy analysis

Cells were seeded in dishes with glass slides and incubated in DMEM containing 10% FBS, then fixed in 4% paraformaldehyde (PFA)/PBS. After that, cells were stained with TOM20 (Abclonal, A19403) at room temperature for 2 h followed by washing in PBS. In the end, the slides were further incubated with fluorescein-conjugated secondary antibodies diluted in blocking buffer for 1 h and mounted with ProLong Diamond Antifade Mountant with DAPI (Beyotime, C1002). Then mitochondrial morphology was determined using Acquire SR software on a DeltaVision OMX SR super-resolution imaging system (GE Healthcare), and the images were further computationally reconstructed and processed with Softworx (GE Healthcare). Images were deconvolved and analyzed by using ImageJ.

## Transmission electron microscopy

Cells were collected and fixed in a solution containing 2.5% glutaraldehyde in 0.1 M sodium cacodylate for 2 h, fixed with 1% OsO4 for 1.5 h, and washed and stained in 3% aqueous uranyl acetate for 1 h. The samples were then washed again, dehydrated with a graded alcohol

series, and embedded in resin. Ultrathin sections were cut, counter-stained with 0.3% lead citrate, and examined on a JEOL transmission electron microscopy. Mitochondrial volume density was calculated as previously described[61].

## Oxygen consumption rate (OCR) measurement

Huh7 cells were seeded at $1.5 \times 10^5$ cells per well into XF96-well plates. Before measurement, cells were washed three times with XF-base medium containing 2 mM Glutamine (pH = $7.4 \pm 0.05$). Then, mito-chondrial poisons (1.5 μM oligomycin, 2 μM FCCP, 0.5 μM rotenone, and 0.5 μM antimycin A) were added at the indicated time points. Oxygen consumption rate were analyzed by Seahorse XFp Wave soft-ware. In the end, cell lysis was harvested with western blot lysis buffer and protein concentration was quantified using Pierce BCA Protein Assay Kit (Thermo Fisher Scientific, 23225). OCR value was normalized to the protein concentration in each well.

## Extracellular oxygen consumption assay

Extracellular Oxygen Consumption Assay was measured using the Oxygen Consumption Kit (Abacm, ab197243) according to the manu-facturer's protocol. Briefly, cells were seeded in a 96-well plate at a density of $4 \times 10^4$ cells/well in 150 μL culture medium. And Extracellular Oxygen Consumption Reagent were added to wells. Data were deter-mined by luminescence (PerkinElmer, Envision). Data were collected from multiple replicate wells for each experiment and normalized to protein concentration.

## Oroboros 2K oxygraph assay

The mitochondrial respiratory function was assayed by measuring oxygen consumption rates (OCRs). We used a real-time high-resolu-tion respirometry (Oxygraph-2k; Oroboros Instruments, Innsbruck, Austria) under a variety of substrate conditions and respiratory states to examine differences in respiratory capacity, collect cells immedi-ately loaded into an Oroboros 2K oxygraph chamber filled with Miro6 buffer equilibrated at 25 °C. In each of these protocols, Oxygen con-sumption rates were measured before and after addition of the fol-lowing sequence of substrates and specific inhibitors: (1) 5 mM pyruvate (Sigma-Aldrich, P4562), 2 mM malate (Sigma-Aldrich, M1000) in flies, followed by 2.5 mM ADP (Sigma-Aldrich, 117105) to determine complex I-driven phosphorylating respiration (CI OXPHOS). (2) 10 mM succinate (Sigma-Aldrich, S2378) to determine the phosphorylating respiration driven by simultaneous activation of complex I and II (CI+II OXPHOS). (3) Titrating concentrations of the mitochondrial uncoupler 0.5 μM CCCP (Sigma-Aldrich, C2759) to reach the maximal, uncoupled respiration (CI+II electron transfer system, ETS). (4) 0.5 μM rotenone (Sigma-Aldrich, R8875) to fully inhibit complex I-driven respiration and measure complex II-driven uncoupled respiration (CII electron transfer system, CII ETS). (5) 2.5 μM Antimycin A (Sigma-Aldrich, A8674) to block mitochondrial respiration at the level of complex III. Residual oxygen consumption was always negligible.

## Luciferase reporter assay

Transcriptional regulation analyses were evaluated using a dual luci-ferase reporter assay system (Promega, E1960), the ETC genes promoter regions were cloned to pGL3-basic vector, or three repeated motifs were cloned to pGL3-promoter vector. Huh7 cells pre-cultured on 24-well plates were transfected with a combination of ZHX2-expressing plas-mid, luciferase reporter plasmid, and pRL-TK. The cells were lysed and collected for analysis of firefly luciferase activity according to the man-ufacturer's protocol and normalized to Renilla luciferase activity.

## Electrophoretic mobility shift (EMSA)

ZHX2 overexpressed and control Huh7 cells were cultured in 100 mm plates with DMEM containing 10% FBS for 48 h. The nuclear extracts

were prepared by using a Nuclear Extraction Kit (Beyotime, P0028). The repeated motif (5′-AGGCTGAGAGGCTGAGAGGCTGAG-3′) were labeled at 3′ (GENEWIZ) and annealed as the probes. Then, the nuclear extracts and probes were incubated according to the protocol of Light Shift Chemiluminescent EMSA kit (Beyotime, GS009). In brief, 20 μL of 1× EMSA/Gel-Shift buffer, 4 μg of nuclear extracts, and 0.2 μM of labeled probes with or without 10 μM of unlabeled competitor oligo-nucleotides were applied and incubated at room temperature for 20 min, respectively. Then DNA-protein complexes were loaded onto a 4% non-denaturing polyacrylamide gel for blotting.

## Biotinylated pull-down assay

The repeated motif (5′-AGGCTGAGAGGCTGAGAGGCTGAG-3′) were biotinylated at 5′ (GENEWIZ) and annealed as the probes. Then, nuclear extracts, biotin-labeled probe, and streptavidin magnetic beads (Thermo Fisher Scientific, 11205D) was incubated, spin column for 2 h on a rocking platform. The spin-down beads were used to collect the binding protein by boiling. The bound protein was loaded in SDS-PAGE, and the signal was detected by enhanced chemiluminescence (ECL) reagent (Millipore, IPVH00010) using the DNR Bio-Imaging Systems. Unlabeled probes incubated with nuclear extracts and streptavidin magnetic beads as negative control.

## Immunoprecipitation and immunoblotting assay

The extraction of proteins was collected using a modified buffer (Beyotime, P0013) from cultured cells and followed by immunopre-cipitation and immunoblotting analyses using corresponding antibodies[62]. For immunoprecipitation, one microgram of protein was incubated with 2 μg antibodies. After overnight incubation at 4 °C, protein G-magnetic beads (Bimake, B23202) were added and incu-bated for another 3 h. Then, the beads were washed by the lysis buffer for four times. Immunocomplexes were analyzed by immunoblotting assay with indicated antibodies.

## Measurement of mtDNA content

Total DNA was extracted from indicated cells by using TIANGEN Genomic DNA Purification Kit (TIANGEN, DP304) according to the manufacturer's instructions. To quantify mtDNA copy number, real-time PCR was performed using a Real-Time PCR system from BioRad C1000 Thermal Cycler CFX96 Real-Time System against the mito-chondrial D-loop region or MT-ND1 as the standard for mtDNA. Beta-2-MicroglobuliN (B2M) was used as the nuclear gene (nDNA) normalizer for the calculation of the mtDNA/nDNA ratio[63,64]. The relative mtDNA content was calculated using the formula: mtDNA content $= 1/2^{\Delta Ct}$, where $\Delta Ct = Ct^{mtDNA} - Ct^{B2M}$.

## ATP, AMP, ATP/AMP and ATP/ADP measurements

ATP and AMP levels, and ATP/AMP ratio were determined using the ATP Determination Kit (Abacm, ab83355) and AMP Determination Kit (Abacm, ab273275) according to the manufacturer's protocol. Briefly, cells were homogenized in lysis buffer supplemented with protease and phosphatase inhibitors. The colorimetric intensity of 50 μL lysis was determined by microplate reader (TECAN). Then, the ATP/AMP ratio was calculated using above measured ATP and AMP levels. ATP/ADP ratio was assessed and calculated by an ADP/ATP Ratio Assay Kit (Abcam, ab65313), 100 μL prepared reaction mix was added in control wells and the background luminescence was read, then 50 μL sample was added and after 2 min the luminescence was read.

## Flow cytometry (FCM) analysis

Cells were incubated with 1 μM Mito Tracker deep Red FM probe (Thermo Fisher Scientific, M22426), 2 μM MitoProbe™ JC-1(Thermo Fisher Scientific, M34152) or 20 μM MitoProbe™ TMRM (Thermo Fisher Scientific, M20036) at 37 °C for 30 min, respectively. After staining was completed, the cells were gently washed three times with

warm PBS, and detached to a single-cell suspension. Analyze the samples by flow cytometry using CytoFLEX S flow cytometer (Beckman).

## Chromatin immunoprecipitation (ChIP) and ChIP-qPCR

Huh7 cells were transfected with ZHX2-HA, and harvested for ChIP assay by using the EZ-Magna ChIP™ A/G Chromatin Immunoprecipitation Kit (Millipore, 17-10086) according to the manufacturer's instructions. Briefly, cells were fixed to extract total DNA. And the DNA was sonicated to 200-1000 bp followed by incubating with antibodies, anti-HA antibody (MBL, M180-3) and rabbit IgG (Santa Cruz, sc-2027). As input, 1/100th of the starting chromatin was used to extract DNA. Specific primers were used for conventional PCR and qPCR for further analysis. Primer for qPCR, see Supplementary Table 1.

## ChIP sequencing (ChIP-seq)

Huh7 cells were transfected with ZHX2-HA, and harvested by using the EZ-Magna ChIP™ A/G Chromatin Immunoprecipitation Kit. DNA sequencing was carried out by Novogene following their standard protocols. The ChIP-seq data in this publication have been deposited in the National Center for Biotechnology Information Sequence Read Archive and are accessible through accession PRJNA798889.

## RNA sequencing and proteomics sequencing

Liver tissue from *Zhx2*-WT and *Zhx2*-KO[hep] mice after 48 h 2/3 PHx were harvested. RNA sequencing and proteomics sequencing were carried out by The Beijing Genomics Institute (BGI) following their standard protocols. Libraries were sequenced on the BGISEQ-500 platform. The RNA-seq data in this publication have been deposited in the National Center for Biotechnology Information Sequence Read Archive and are accessible through accession PRJNA754419. The iTRAQ proteomics data have been deposited to the ProteomeXchange Consortium via the PRIDE partner repository with the dataset identifier PXD027897.

## Gene expression studies

Gene expression data were obtained from GEO omnibus (https://www.ncbi.nlm.nih.gov/geo/), including liver regeneration (GSE63742), nonalcoholic fatty liver disease (NAFLD) (GSE49541) and liver failure (GSE168049). Gene expression of liver hepatocellular carcinoma data was downloaded from Cbioportal (https://www.cbioportal.org/). These data were used to analyze the correlation of ZHX2 with ETC genes. Peak browsing and representative snapshots capturing were performed using the Integrative Genomics Viewer (IGV; IGV2.4.10, Broad Institute).

## Protein half-life assay

After transfection, 500 µg/mL cycloheximide (Selleck, S7418) was added into the medium to stop protein synthesis and maintained for 0, 1, 2, 3, 4 and 5 h, respectively. Cell lysis was collected to analyze the protein levels of PGC-1α by western blot. Quantification of expression of PGC-1α protein was normalized to β-actin under different time points by ImageJ software.

## Ubiquitination analysis

Huh7 cells were co-transfected with indicated plasmids. After 24 h, cells were added with 10 µM MG132 (Sigma-Aldrich, M7449) for 6 h, then were collected for western blot. Ub antibody was used to determine the ubiquitination of PGC-1α.

## Primary hepatocyte isolation

Primary hepatocytes were isolated from 8-week-old mice by a two-step collagenase perfusion method. HBSS (Thermo Fisher Scientific, 14175095) was used to perfuse the liver at 10 mL/min speed until the liver turned pale. Afterward, the liver was perfused with HBSS digestion buffer (30 mg/100 mL collagenase IV (Worthington, LS004189), 2

tablet/100 mL protease inhibitor) at 15 mL/min speed for 18 min. After sequential flows, cells were smashed through 100 µm strainer and washed with Williams' Medium E (Thermo Fisher Scientific, 22551089). Hepatocytes were isolated by density gradient centrifugation using percoll (Pharmacia, Sweden). Primary hepatocytes were cultured for the following experiments.

## Liver organoid culture and lentivirus infection

Human liver organoids were established from single cells from colleganse-digested liver biopsies and cultured in the organoid culture medium. Organoids at passage 8 were removed from Matrigel (Bio-Techne, BME001) and digested with trypsin into single cells. Cells were resuspended in 1 mL of lentivirus solution with 2 µg/mL Polybrene (GeneChem, REVG0001) and subsequently plated in 24-well wells with centrifugation at 500 g for 1 h at room temperature. Then organoid cells were incubated at 37 °C for 4 h and reseeded in the mixture of Matrigel and culture medium. RNA, DNA and proteins were collected from organoids 48 h after infection.

## Statistical analysis

Flow cytometry data was analyzed using FlowJo software. Statistical analysis was carried out with GraphPad Prism 8 software. $N$ represents the number of mice per group used in each experiment, the number of biologically independent experiments performed with cells unless specified otherwise. For single comparisons, the statistical significance was analyzed using Student's $t$-test. For multiple means of comparison, a two-way analysis of variance coupled with Tukey was performed. Data analysis. $p$ value of less than 0.05 was considered statistically significant.

## Reporting summary

Further information on research design is available in the Nature Portfolio Reporting Summary linked to this article.

## Data availability

The RNA-seq data generated in this study have been deposited in the National Center for Biotechnology Information Sequence Read Archive and are accessible through accession PRJNA754419 and the ChIP-seq data generated in this study have been deposited in the National Center for Biotechnology Information Sequence Read Archive and are accessible through accession PRJNA798889. The proteomics data have been deposited to the ProteomeXchange Consortium via the PRIDE partner repository with the dataset identifier PXD027897. All other data supporting the findings of this study are available within the article and its supplementary information files or from the corresponding author upon reasonable request. Source Data are provided with this paper.

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

## Acknowledgements

This work was supported by Taishan Scholarship (No. tspd20181201 (C.M.)), National Natural Science Foundation of China (Key project 81830017 (C.M), 81972819 (X.Y.)), Shandong Natural Science Foundation (ZR2020ZD12 (C.M.), ZR2020YQ57 (X.Y.), ZR2023QH238 (Y.Z.)) and Collaborative Innovation Centre of Technology and Equipment for Biological Diagnosis and Therapy in Universities of Shandong. Key Research and Development Program of Shandong (2019GSF108238). We appreciate Professor Lei Sun (Department of Pathology, Beijing Ditan Hospital Capital Medical University) for the support on clinical characteristics and pathological materials for 15 patients with drug-induced liver injury. We thank Dr. Brett T Spear (University of Kentucky College of Medicine) for gifting hepatocyte-specific *Zhx2* deficient mice (*Zhx2*-KO^hep). We thank Translational Medicine Core Facility of Shandong University for consultation and instrument availability that supported this work.

## Author contributions

Conceptualization: C.M., Xuetian Yue, and Y.Z. Methodology: Y.Z., Xuetian Yue, Zhuanchang Wu, Xiaohui Zhang, Zehua Wang, and H.H. Formal analysis: Xuetian Yue, S.T., X.S., T.W. Investigation: Y.Z., Xiangguo Yu, L.W., Zhuanchang Wu, P.X., T.W., and Xiaodong Zhang. Writing—original draft: C.M., Xuetian Yue, N.L., S.L., and Y.Z. Writing—review and editing: all authors. Funding acquisition: C.M., Y.Z. and Xuetian Yue. Resources: Y.Z., H.H., Y.F., Zhuanchang Wu, and C.L. Supervision: L.G., X.L., and C.M.

## Competing interests

The authors declare no competing interests.
