## [Peer Review File · Nature Communications]

ZHX2 emerges as a negative regulator of mitochondrial oxidative phosphorylation during acute liver injuryREVIEWER COMMENTS

Reviewer #1 (Remarks to the Author):

In the present manuscript, Zhang et al uncover ZHX2 as a regulator of mitochondrial oxidative phosphorylation in hepatocytes. By an elegant combination of biochemical, genetic and bioinformatic approaches, the authors characterized ZHX2 a key regulator of the expression of components of electron transport chain. Consequently, the induction of ZHX2 in models of acute liver injury (e.g. CCl₄) or regeneration (partial hepatectomy) mediates mitochondrial functional impairment, which is reversed upon genetic ZHX2 downregulation. Similar results are shown in a cohort of DILI patients which exhibit increased expression of liver ZHX2. The mechanism of action of ZHX2 in the regulation of mitochondrial respiratory chain involves PGC1 α ubiquitination and subsequent degradation via FBXW7.

COMMENTS

The manuscript is well developed and carefully executed using a wide range of approaches, including biochemical, genetic and bioinformatic and the conclusions are largely supported by the data.

However, my enthusiasm is somewhat reduced by the implications/relevance of the findings.

1. Authors argue that ZHX2 is a novel regulator of the mitochondrial oxidative phosphorylation, as conveyed in the title. It should be emphasized that other regulators of mitochondrial oxidative phosphorylation has been already described, such as MCJ (Hatle et al., Mol Cell Biol 2013; Champagne et al. Immunity 2016). Thus, ZHX2 emerges as another regulator of mitochondrial oxphos, which should be clarified in title and abstract and in the manuscript.

2. The impact of ZHX2 is mainly tested in acute conditions, such as in partial hepatectomy and acute liver injury by CCl₄. I think the role of ZHX2 should be expanded in a metabolic disorder, like NASH.

3. This new context of research in NASH would be very interesting as the contribution of alterations of mitochondrial function to disease progression from steatosis to obesity and hepatosteatosis is controversial. Cumulative genetic evidence, including but not limited to the hepatocyte-specific deletion of AIF (Pospisilik et al, Cell 2007) indicated that the loss of mitochondrial oxidative phosphorylation activity protected against diet induced steatosis, insulin resistance and o NASH progression.

4. Related to the previous point the role of metformin, a mitochondrial complex I inhibitor, should be expanded in a metabolic liver disorder, as it is a useful drug for T2D, and insulin resistance.

5. The impact of ZHX2 downregulation in CCl₄-mediated liver injury is difficult to appreciate. The ALT/AST values of Fig 2E appear quiet modest, and contrast to the marked difference shown in the H&E staining shown in Fig 2F-G.

6. Quality of TEM pictures shown in Fig 3F is questionable as the images are difficult to see any detail.

7. In the cohort of DILI patients, it should be clarified what kind of drugs were responsible for the liver damage in this population.

Reviewer #2 (Remarks to the Author):

This study investigated the role of ZHX2 in mitochondrial oxidative phosphorylation and liver regeneration. This study showed that ZHX2 inhibits OXPHOS and suppresses ATP production. Conversely, knockout of ZHX2 in mouse liver promotes liver regeneration after partial hepatectomy and CCl₄-induced liver injury through ATP production. In addition, the authors showed that ZHX2 directly represses transcription of several ETC-related genes and also promotes ubiquitination of PGC1 α via increased expression of the ubiquitin ligase FBXW7. The promotion of liver regeneration in ZHX2 KO mice was cancelled by PGC1 α inhibitors, suggesting that the promotion of liver regeneration is mediated by an increase in PGC1 α . This manuscript contains several novel findings and the experimental design is appropriate to test their hypothesis. I have some comments as follows.

Comments

1. The authors state that ZHX2 promotes ubiquitination of PGC1 α via the ubiquitin ligase FBXW7. However, if ZHX2 regulates FBXW7 expression, a broader range of proteins may be affected. For example, since FBXW7 is known to ubiquitinate c-Myc and cyclin E, the decrease in FBXW7 by ZHX2 knockout may increase the expression of cell cycle-related proteins, which could contribute to the promotion of liver regeneration. Furthermore, the authors previously reported that ZHX2 inhibits HCC cell proliferation by direct inhibition of cyclin A and E expression (*Gastroenterology*. 2012 Jun;142(7):1559-70.e2.). Therefore, promotion of liver regeneration by ZHX2 knockout may be caused by acceleration of cell cycle rather than ATP production. The authors should discuss this point.
2. How does ZHX2 regulate the expression of FBXW7?
3. It has been reported that ZHX2 positively regulates PTEN expression and knockout of ZHX2 induces activation of PI3K-Akt-mTOR pathway (*Hepatology*. 2022 Apr;75(4):939-954.). However, PTEN expression or activation status of PI3K-Akt-mTOR pathway in ZHX2 knockout mice was not assessed in this paper. Have the authors looked into that? Activation of mTOR pathway could explain the increased hepatocyte proliferation and ATP production. The story of this study is well-designed, but somewhat complex. Most of phenotypes of ZHX2 knockout mice might be explained by mTOR activation. So, the authors should carefully assess this issue.

Reviewer #3 (Remarks to the Author):

General comments:

The molecular mechanisms governing hepatocyte regeneration following injury are of interests to both academic and clinical communities. In this study, Zhang et al. present their findings, based on a curated candidate approach that the zinc-finger transcription factor Zhx2 is a major regulator of hepatocyte regeneration following injury in both mouse models and human clinical data. They provide evidence that Zhx2 negatively regulates oxidative phosphorylation and propose that this happens through negative regulation of OXPHOS gene expression as well as stability of PGC1- α . I think the paper raises an interesting finding of potential interest to a wide audience that Zhx2 is a modifier in the ability of liver to

regenerate following injury, and correlatively in liver cancer progression. However, the manuscript in its current form has a few major shortcomings in approach and data interpretation that require further investigation before the conclusions made in the paper are warranted.

First, it is unclear if Zhx2 status has a lasting effect on liver function in the various models of liver injury used in the paper beyond a transient effect seen in regeneration. For instance, is there a change in liver fibrosis or liver function beyond the acute phase? In the context of liver cancer, the effect of Zhx2 on OXPHOS is correlative at best and seems at odds with the previous findings from the same group that Zhx2 is a tumor suppressor working through the suppression of LPL (PMID: 31740790). While the authors use that as an entry way to suggest that Zhx2 might act directly on the ETC, another possibility is that the effects observed are downstream of changes in LPL and lipid metabolism. With regards to liver regeneration, the authors claim that OXPHOS status is a major determinant of hepatic regeneration following injury, and hence that's why Zhx2's function in regulating OXPHOS in might be so important. But what is the evidence that OXPHOS status is key to hepatic regeneration in the models used and if so, what is the mechanism? For instance, is aspartate metabolism in cells with high flux through the TCA (and ETC) crucial for hepatocyte regeneration? Without first demonstrating this point, it seems presumptive to implicate Zhx2's role in OXPHOS as the primary reason for the observed differences in regeneration. It will also be useful to see if there are any sub-clinical GWAS hits that provide evidence at the genetic level for a clinically relevant involvement of Zhx2 in liver disease in humans.

Assuming the above point is proven, the proposed model of how Zhx2 antagonizes ETC OXPHOS is currently inconsistent. In Figure 5, authors propose that OXPHOS suppression is due to the activity of ZHX2 as a transcriptional repressor, occupying genes in the ETC. These data are not so convincing due to the small number of genes found to be direct targets. However in Figure 6, authors switch to a different mechanism, where Zhx2 appears to potentiate the expression of genes involved in ubiquitinylation UPS system, including FBXW7, leading to increased PGC1a ubiquitinylation and degradation. In this respect, it is unclear if Zhx2 is acting directly as a transcriptional activator or via other mechanisms, and if so, what other mechanisms that are downstream of its earlier proposed role as a transcriptional repressor.

In figure 4, the use of Metformin and FCCP as methods to demonstrate that Zhx2 is acting on OXPHOS to affect liver regeneration are misleading. Metformin has pleiotropic effects, including activation of AMPK which should lead to a general augmentation of OXPHOS biogenesis in addition to the specific inhibition of Complex I. FCCP administration causes depolarization and is generally toxic. To conclude that FCCP reverses the regenerative effect Zhx2 KO settings, thereby proving that Zhx2 acts via OXPHOS regulation, is really misleading. Unless the authors include a negative control whereby FCCP administration does not affect the regenerative effects of another gene (not related to OXPHOS), I would interpret the findings in Figure 4 to be caused by nonspecific toxicity of FCCP.

Specific comments:

- Textual comments:

o Grammatical and syntax errors throughout the manuscript that need to be fixed, especially in the introduction.

o Descriptions of "ectopic expression", "augmented expression" etc, is difficult to follow if ZHX2 in the system is overexpressed. Best to keep to one definition of overexpression.

- Figure 2: Chip-SEQ data presented are restricted to OXPHOS genes. What about other classes of genes that are affected in Zhx2 KO that impinge on hepatocyte proliferation? Does Zhx2 directly affect transcription of cyclin A2/B1 etc and increase cell proliferation? Do

they have Zhx2 binding sites? This is also observed in Fig S3C where cell cycle genes are upregulated in Zhx2-KO mice. Figure 2D: data in plot is reverse compared to other panels.

- Figure 3:

o Figures and legend do not correspond to each other.

o What are the differences at baseline for control vs Zhx2 KO without injury? Are these pathways enriched even further upon injury?

o Fig 3A-B: Gene sets are very general pathways. Are there specific mitochondrial pathways that are enriched? Fig 3B: "Energy metabolism" is as enriched as other pathways (lipid, amino acid metabolism).

o Figure S3D: fold change >1.2 seems very lenient, are there pathways enriched when cut off is more stringent?

o Results line 228: expression correlative and only a small number of genes (4) were shown to be direct repression targets. As such, authors should be careful to conclude ZHX2 as a direct inhibitor of OXPHOS.

- Figure 4 :

o Figure 4A: Metformin can already disrupt membrane potential. JC-1 might not be the best readout for mitochondrial health here.

o Figure 4D/4E/4F/4G: The observed increases in gene expression are quite small, albeit not statistically significant in KO upon metformin treatment. Are there other mitochondria-independent pathways that help recovery?

- Figure 5B: What about other ETC proteins? Are other mitochondrial pathways also affected since mitochondrial mass increases?

- Figure 6D: overexpression levels not compatible to controls, especially in CQ panel

- Figure 7D: only a few ETC genes are affected by Zhx2 direct binding, and a few are affected by PGC1a instability. Are the reduction in expression of these genes enough for reduction of mitochondrial mass?

- Figure 8D: Is Tom20 PGC1a controlled? Are there other mitochondrial markers apart from TOM20 that are coexpressed with Zhx2 upon DILI?

Responses to Reviewers

Thank you for your insightful comments and suggestions on our work. The manuscript has been carefully revised according to your comments. We believe that the revision has significantly improved our manuscript. Our point-to-point responses to your specific comments are as follows.

Response to the comments of Reviewer 1:

Reviewer #1 (Remarks to the Author):

In the present manuscript, Zhang et al uncover ZHX2 as a regulator of mitochondrial oxidative phosphorylation in hepatocytes. By an elegant combination of biochemical, genetic and bioinformatic approaches, the authors characterized ZHX2 a key regulator of the expression of components of electron transport chain. Consequently, the induction of ZHX2 in models of acute liver injury (e.g. CCl4) or regeneration (partial hepatectomy) mediates mitochondrial functional impairment, which is reversed upon genetic ZHX2 downregulation. Similar results are shown in a cohort of DILI patients which exhibit increased expression of liver ZHX2. The mechanism of action of ZHX2 in the regulation of mitochondrial respiratory chain involves PGC1a ubiquitination and subsequent degradation via FBXW7.

COMMENTS

The manuscript is well developed and carefully executed using a wide range of approaches, including biochemical, genetic and bioinformatic and the conclusions are largely supported by the data. However, my enthusiasm is somewhat reduced by the implications/relevance of the findings.

1. Authors argue that ZHX2 is a novel regulator of the mitochondrial oxidative phosphorylation, as conveyed in the title. It should be emphasized that other regulators of mitochondrial oxidative phosphorylation has been already described, such as MCJ (Hatle et al., Mol Cell Biol 2013; Champagne et al. Immunity 2016). Thus, ZHX2 emerges as another regulator of mitochondrial oxphos, which should be clarified in title and abstract

and in the manuscript.

Response: As suggested, we have changed the title to “ZHX2 emerges as a regulator of mitochondrial oxidative phosphorylation in hepatocytes” (**page 1 , line 1**) and have emphasized that other regulators of mitochondrial oxidative phosphorylation has been reported during progression of liver diseases in the abstract, introduction and discussion of the revised manuscript (**page 2, line 41-42, 50; page 3, line 79-83; page 17, line 387-388**). Furthermore, we have cited references about MCJ in the introduction of the revised manuscript (**page 4, line 91-94**).

2. The impact of ZHX2 is mainly tested in acute conditions, such as in partial hepatectomy and acute liver injury by CCl4. I think the role of ZHX2 should be expanded in a metabolic disorder, like NASH.

Response: Thanks for this suggestion. Indeed, previous studies including ours had been reported that ZHX2 involves in the progression of NASH. Our data showed that overexpression of ZHX2 prevents NAFLD progression by preventing hepatic lipid accumulation in HFD mice (**Wu et al., Cell Death and Differentiation 2020, PMID: 31740790**). Recently, Zhao et al. showed that hepatocyte-specific overexpression of ZHX2 mitigates HFHC-induced NASH through transactivation of PTEN (**Zhao et al., Hepatology 2022, PMID: 34545586**). These findings have been cited in our revised manuscript (**page 10, line 217-218**).

3. This new context of research in NASH would be very interesting as the contribution of alterations of mitochondrial function to disease progression from steatosis to obesity and hepatosteatosis is controversial. Cumulative genetic evidence, including but not limited to the hepatocyte-specific deletion of AIF (Pospisilik et al, Cell 2007) indicated that the loss of mitochondrial oxidative phosphorylation activity protected against diet induced steatosis, insulin resistance and o NASH progression.

Response: Thanks for these insightful comments. We agree with the comments of reviewer that the contribution of alterations of mitochondrial function to disease progression from steatosis to obesity and hepatosteatosis is controversial. Mitochondrial dysfunction is

manifest as a variety of disorders with clinical outcomes largely dependent on the magnitude and tissue distribution of the impairment (**Nick Lane, Nature 2006, PMID: 16572142; Douglas C Wallace, Science 1999, PMID: 10066162**). Hepatocyte-specific deletion of inner-mitochondrial membrane-associated protein AIF leads to the loss of mitochondrial oxidative phosphorylation which in turn counteracts the development of diet induced steatosis, insulin resistance and obesity (**Pospisilik et al., Cell 2007, PMID: 17981116**). Similarly, here our results showed that ZHX2 levels are negatively associated with mitochondrial OXPHOS (**Fig.3 and Supplementary Fig.1A (RNA-seq data of Zhao's paper, PMID: 34545586)**). Notably, previous research including ours showed that hepatocyte-specific overexpression of ZHX2 retards NAFLD and NASH progression (**Wu et al., Cell Death and Differentiation 2020, PMID: 31740790; Zhao et al., Hepatology 2022, PMID: 34545586**). These studies suggest ZHX2 as a regulator of mitochondrial OXPHOS, and provide new evidence that loss of mitochondrial oxidative phosphorylation activity in hepatocytes protected against diet-induced NASH progression. Whereas, the function and underlying mechanisms of mitochondrial oxidative phosphorylation in metabolic disorders need further investigation.

4. Related to the previous point the role of metformin, a mitochondrial complex I inhibitor, should be expanded in a metabolic liver disorder, as it is a useful drug for T2D, and insulin resistance.

Response: Thanks for this constructive suggestion. One of main interests of our lab is studying the biological function of ZHX2 in pathological and physiological conditions, including T2D. We have detected the expression of ZHX2 in liver and pancreas of db/db mice in our ongoing projects. The results showed that *Zhx2* mRNA levels were reduced in liver and pancreas of db/db mice compared with those in the control mice (**Ding et al., iScience 2023, PMID: 37275527 and Reviewer Only Figure 1A**), indicating the potential role of ZHX2 in regulating the progression of T2D. In fact, our recent study showed that β -cell-specific *Zhx2* knockout is associated with reduced β cell mass, insulin secretion and glucose intolerance in both HFD-induced T2D mice and db/db mice (**Ding et al., iScience 2023, PMID: 37275527**). Furthermore, we also tried to decipher the role of hepatic ZHX2

in regulating T2D progression. Glucose tolerance test (GTT) results showed that knockdown of *Zhx2* in db/db mice liver induced glucose intolerance. As shown in **Reviewer Only Figure 1B and 1C**, *Zhx2* deficiency in hepatocyte impairs glucose homeostasis. In addition, metformin treatment reversed the effect of *Zhx2* knockdown on glucose tolerance (**Reviewer Only Figure 1C**). This part of work is still ongoing. Hopefully, we will give the audience the clear answer in the near future.

Review Only Figure 1. ZHX2 involves in the progression of T2D. (A) The mRNA levels of *Zhx2* were detected in liver of db/db and control mice by RT-qPCR. (B-C) AAV-shZhx2 were delivered to db/db mice *via* tail vein injection then treated with metformin, following by glucose tolerance test (GTT). Liver-specific knockdown of ZHX2 was confirmed by western blot (B). For GTT, ZHX2 liver-specific knockdown db/db mice were fasted for 18 h and pretreat with metformin for 8 h, followed by intraperitoneally injected glucose (2 g/kg body weight). Blood glucose levels were measured at indicated times (left panel) and area under the curve (AUC) (right panel) were presented (C).

5. The impact of ZHX2 downregulation in CCl4-mediated liver injury is difficult to appreciate. The ALT/AST values of Fig 2E appear quiet modest, and contrast to the marked difference shown in the H&E staining shown in Fig 2F-G.

Response: Thanks for these careful review and kind remaindering. We re-analyzed the serum ALT and AST using alanine transaminase activity assay (Abacm, ab105134) and aspartate aminotransferase activity assay (Abacm, ab105135) which has higher sensitivity. The update data was presented in the revised manuscript (**Fig.2E**), which showed the clear difference.

6. Quality of TEM pictures shown in Fig 3F is questionable as the images are difficult to see any detail.

Response: In order to show the detail of mitochondria, we have selected one mitochondrion in the field to zoom in. As shown in **Fig.3F**, mitochondrion showed an elongated morphology in *Zhx2*-KO^{hep} mice liver, and the number of mitochondrial cristae was increased in *Zhx2*-KO^{hep} mice liver cells compared to that of *Zhx2*-WT mice.

7. In the cohort of DILI patients, it should be clarified what kind of drugs were responsible for the live damage in this population.

Response: Thanks for this very useful suggestion. In the cohort of DILI patients, liver damage was caused by drugs, such as angiotensin-converting enzyme inhibitor, antidiabetic agents, anticonvulsant drugs, dietary supplements, Chinese herbal medicine, antibiotics, acetaminophen, antipsychotic agents, and antihistamines. We have updated this information in the revised manuscript (**page 20, line 446-451**).

Response to the comments of Reviewer 2:

Reviewer #2 (Remarks to the Author):

This study investigated the role of ZHX2 in mitochondrial oxidative phosphorylation and liver regeneration. This study showed that ZHX2 inhibits OXPHOS and suppresses ATP production. Conversely, knockout of ZHX2 in mouse liver promotes liver regeneration after partial hepatectomy and CCl4-induced liver injury through ATP production. In addition, the authors showed that ZHX2 directly represses transcription of several ETC-related genes and also promotes ubiquitination of PGC1a via increased expression of the ubiquitin ligase FBXW7. The promotion of liver regeneration in ZHX2 KO mice was cancelled by PGC1a inhibitors, suggesting that the promotion of liver regeneration is mediated by an increase in PGC1a. This manuscript contains several novel findings and the experimental design is appropriate to test their hypothesis. I have some comments as follows.

Comments

1. The authors state that ZHX2 promotes ubiquitination of PGC1a via the ubiquitin ligase FBXW7. However, if ZHX2 regulates FBXW7 expression, a broader range of proteins may

be affected. For example, since FBXW7 is known to ubiquitinate c-Myc and cyclin E, the decrease in FBXW7 by ZHX2 knockout may increase the expression of cell cycle-related proteins, which could contribute to the promotion of liver regeneration. Furthermore, the authors previously reported that ZHX2 inhibits HCC cell proliferation by direct inhibition of cyclin A and E expression (*Gastroenterology*. 2012 Jun;142(7):1559-70.e2.). Therefore, promotion of liver regeneration by ZHX2 knockout may be caused by acceleration of cell cycle rather than ATP production. The authors should discuss this point.

Response: Thanks for these helpful comments. Indeed, many biological processes and signaling pathways involve in liver regeneration. Here, we identified that mitochondrial oxidative phosphorylation is one of the most obviously changed biological processes in *Zhx2*-KO^{hep} mice liver during liver regeneration. Furthermore, our data demonstrated that knockdown or knockout of ZHX2 increased mitochondrial OXPHOS and ATP production to promote liver regeneration in both *in vitro* and *in vivo* experiments. We do agree that cell cycle regulation plays important role in liver regeneration. Previous reports including ours demonstrated that ZHX2 and FBXW7 regulated expression of cell cycle-related genes, such as *cyclin A*, *cyclin E* and *c-Myc* (Yue *et al.*, *Gastroenterology* 2012, PMID: 22406477; Kitagawa *et al*, *Oncogene* 2009, PMID: 19421138; King *et al*, *Cell* 2013, PMID: 23791182). Therefore, cell cycle might contribute to the effect of ZHX2 in liver regeneration. We have discussed these possibilities in revised manuscript (page 18, line 402-404; page 19, line 425-428).

2. How does ZHX2 regulate the expression of FBXW7?

Response: Thanks for this insightful question. To decipher the molecular mechanism of ZHX2 in regulating FBXW7 expression, the promoter of FBXW7 were cloned to luciferase reporter plasmid. Luciferase reporter assays showed that luciferase activity was increased in ZHX2 transfected Huh7 cells compared with that of control cells (Supplementary Fig.6H), indicating ZHX2 promotes transcription of FBXW7. Then, we further analyzed our ChIP-seq data and identified one of ZHX2 binding-motif located on the promoter of FBXW7 (-1045~-1038 bp) (Supplementary Fig.6I). Using specific primers targeting to this ZHX2-binding motif on FBXW7 promoter, ChIP assays confirmed the enrichment of ZHX2 on

FBXW7 promoter (**Supplementary Fig.6J**). All these data suggest that ZHX2 binds to FBXW7 for transactivation, which is consistent with previous study reporting ZHX2 as a transcriptional activator (**Zhang et al., Science 2018, PMID:30026228**). This has been included in the revision (**Supplementary Fig.6H-6J**)(page 14, line 317-319).

3. It has been reported that ZHX2 positively regulates PTEN expression and knockout of ZHX2 induces activation of PI3K-Akt-mTOR pathway (Hepatology. 2022 Apr;75(4):939-954.). However, PTEN expression or activation status of PI3K-Akt-mTOR pathway in ZHX2 knockout mice was not assessed in this paper. Have the authors looked into that? Activation of mTOR pathway could explain the increased hepatocyte proliferation and ATP production. The story of this study is well-designed, but somewhat complex. Most of phenotypes of ZHX2 knockout mice might be explained by mTOR activation. So, the authors should carefully assess this issue.

Response: Thanks for the comments. As the reviewer pointed out, ZHX2 positively regulates PTEN expression to inactivate PI3K-Akt-mTOR pathway (**Zhao et al., Hepatology. 2022, PMID: 34545586**), which might also involve in ZHX2 mediated regulation of liver regeneration. Since our RNA-seq data showed that mitochondrial OXPHOS is the most changed pathway in *Zhx2*-KO^{hep} mice liver after 2/3 PHx, we focused on this pathway in this manuscript. Our data confirmed that loss of ZHX2 accelerates liver recovery after different kinds of injury by enhancing mitochondrial OXPHOS. To assess the extent to which PI3K-Akt-mTOR pathway is involved in ZHX2 mediated regulation of liver regeneration, we performed the following analysis and experiments. First, we did GSEA analysis and compared the expression of PI3K-Akt-mTOR pathway. Although GSEA did not show significant enrichment of PI3K-Akt-mTOR pathway in *Zhx2*-KO^{hep} mice liver at 48 h after 2/3 PHx (**Reviewer only Figure 2A**), the hepatic expression of many genes involving in PI3K-Akt-mTOR pathway were up-regulated in *Zhx2*-KO^{hep} mice liver at 48h after 2/3 PHx (**Reviewer only Figure 2B**). Furthermore, western blot showed that phosphorylation of mTOR (S2448) in *Zhx2*-KO^{hep} were higher than that in *Zhx2*-WT mice liver at 48h after 2/3 PHx, and this enhanced phosphorylation was abolished by rapamycin treatment (**Reviewer only Figure 2C**), which is consistent with the previous study (**Zhao**

et al., Hepatology. 2022, PMID: 34545586). Importantly, although rapamycin treatment inhibited liver regeneration at 48h after 2/3 PHx, it could not fully block the accelerating liver regeneration in *Zhx2*-KO mice (**Reviewer only Figure 2D**). These data demonstrate that mTOR signaling plays a role in ZHX2 regulating liver regeneration, but it cannot fully explain the phenotype of ZHX2 knockout mice. We have included this in the discussion of revised manuscript (**page 19, line 429-434**).

Reviewer only Figure 2. Activation of mTOR pathway in *Zhx2*-KO^{hep} mice liver during liver regeneration. (A) GSEA was performed to analyze the genes with differential mRNA levels in *Zhx2*-KO^{hep} and *Zhx2*-WT mice at 48 h after 2/3 PHx. (B) A heatmap from A based on the expression of genes in PI3K-Akt-mTOR pathway. (C) The phosphorylation mTOR at S2448 were detected in both *Zhx2*-KO^{hep} and *Zhx2*-WT mice with or without rapamycin treatment by western blot. The levels of p-mTOR/mTOR were calculated. (D) The liver/body ratios were calculated in *Zhx2*-KO^{hep} and *Zhx2*-WT mice with or without rapamycin treatment at 48 h after 2/3 PHx.

Response to the comments of Reviewer 3:

Reviewer #3 (Remarks to the Author):

General comments:

The molecular mechanisms governing hepatocyte regeneration following injury are of interests to both academic and clinical communities. In this study, Zhang et al. present their

findings, based on a curated candidate approach that the zinc-finger transcription factor *Zhx2* is a major regulator of hepatocyte regeneration following injury in both mouse models and human clinical data. They provide evidence that *Zhx2* negatively regulates oxidative phosphorylation and propose that this happens through negative regulation of OXPHOS gene expression as well as stability of PGC1-alpha. I think the paper raises an interesting finding of potential interest to a wide audience that *Zhx2* is a modifier in the ability of liver to regenerate following injury, and correlatively in liver cancer progression. However, the manuscript in its current form has a few major shortcomings in approach and data interpretation that require further investigation before the conclusions made in the paper are warranted.

First, it is unclear if *Zhx2* status has a lasting effect on liver function in the various models of liver injury used in the paper beyond a transient effect seen in regeneration. For instance, is there a change in liver fibrosis or liver function beyond the acute phase? In the context of liver cancer, the effect of *Zhx2* on OXPHOS is correlative at best and seems at odds with the previous findings from the same group that *Zhx2* is a tumor suppressor working through the suppression of LPL (PMID: 31740790). While the authors use that as an entry way to suggest that *Zhx2* might act directly on the ETC, another possibility is that the effects observed are downstream of changes in LPL and lipid metabolism. With regards to liver regeneration, the authors claim that OXPHOS status is a major determinant of hepatic regeneration following injury, and hence that's why *Zhx2*'s function in regulating OXPHOS in might be so important. But what is the evidence that OXPHOS status is key to hepatic regeneration in the models used and if so, what is the mechanism? For instance, is aspartate metabolism in cells with high flux through the TCA (and ETC) crucial for hepatocyte regeneration? Without first demonstrating this point, it seems presumptive to implicate *Zhx2*'s role in OXPHOS as the primary reason for the observed differences in regeneration. It will also be useful to see if there are any sub-clinical GWAS hits that provide evidence at the genetic level for a clinically relevant involvement of *Zhx2* in liver disease in humans.

Response: Thanks for these reasonable and insightful comments. As the last effect of ZHX2 is concerned, previous work including ours showed that ZHX2 plays important role

in NAFLD, NASH, T2D and NAFLD-HCC, all of which support the lasting effect of ZHX2 (Wu *et al.*, *Cell Death Differ* 2020, PMID: 31740790; Zhao *et al.*, *Hepatology* 2022, PMID: 34545586; Ding *et al.*, *iScience* 2023, PMID: 37275527; Reviewer Only Figure 1). In addition, we also had some preliminary data about the role of ZHX2 in liver fibrosis. We used low-dose injection of CCl₄ to induce liver fibrosis in hepatocyte-specific *Zhx2* knockout mice and littermate controls. The results showed that hepatic deficiency of *Zhx2* aggravates CCl₄-induced liver fibrosis, displaying as increased Sirius red staining and Masson staining (Reviewer Only Figure 3). We hope we can introduce this ongoing work to the audience in the near future. We have included some of these findings in the revised manuscript (page 10, line 217-218).

Reviewer only Figure 3. The role of ZHX2 in CCl₄-induced liver fibrosis. 25% CCl₄ (5 mL/kg) were used to treat *Zhx2*-KO^{hep} and *Zhx2*-WT mice for two months. **(A)** ZHX2 protein levels were detected in *Zhx2*-WT and *Zhx2*-KO^{hep} mice liver tissues by western blot. **(B)** Sirius red staining under the light microscope indicates total collagen deposits. Representative images and quantitative data were presented. **(C)** Masson staining for collagen in liver biopsy specimens from CCl₄-treated mice. Representative images and quantitative data were presented.

Emerging evidence demonstrate that mitochondria play a central role in the liver regeneration (Verma *et al.*, *J Hepatol* 2022, PMID: 35777586; Caldez *et al.*, *Dev Cell* 2018, PMID: 30344111; Solhi *et al.*, *Trends Endocrinol Metab* 2021, PMID: 34304970). After hepatectomy, mitochondrial OXPHOS is required to supply hepatocytes with increasing amounts of adenosine triphosphate (ATP), which is needed to fuel biosynthesis of cell components and progression of cell cycle (G Y Minuk, *Can J Gastroenterol* 2003,

PMID: 12915914). Improvement of mitochondrial biogenesis could increase mitochondrial OXPHOS to promote liver regeneration after PHx (**Li et al., Life Sci 2018, PMID: 30473024**). Since our RNA-seq data showed mitochondrial OXPHOS is the most significant changed pathway in *Zhx2*-KO^{hep} mice liver at 48 h after 2/3 PHx. We focused on the role of ZHX2 in regulating mitochondrial OXPHOS in the present manuscript. Mechanistically, ZHX2 suppresses ETC genes expression through PGC-1 α -dependent and -independent manners to inhibit mitochondrial OXPHOS during liver regeneration. Although aspartate metabolism affects TCA flux, our RNA-seq data did not show any significant changes related to this pathway (**Reviewer Only Figure 4**). To emphasize the importance of mitochondrial OXPHOS in liver regeneration, we had updated findings of mitochondrial OXPHOS in hepatic regeneration in the introduction of the revised manuscript (**page 3, line 79-83**).

Our previous work reported that ZHX2 transcriptionally represses LPL to decrease exogenous lipids uptake and ATP generation to retard NAFLD progression (**Wu et al., Cell Death Differ 2020, PMID: 31740790**). Since fatty acids are important fuels for mitochondrial ATP generation, ZHX2-LPL axis might also play a role in ZHX2-mediated regulation of liver regeneration. We have included this in the discussion of revised manuscript (**page 19, line 429-434**).

It is worth to note that mitochondria also play important roles in cancer. Multiple aspects of mitochondrial biology, including bioenergetics, biogenesis and turnover, fission and fusion dynamics, contribute to transformation, cell death susceptibility, oxidative stress regulation, metabolism, and signaling in cancer development (**Vyas et al., Cell 2016, PMID: 27471965**). Therefore, ZHX2 might exert its tumor suppressor function through downregulation of mitochondrial function, which need further studies.

Reviewer only Figure 4. The change of aspartate metabolism was examined between

Zhx2-KO^{hep} and Zhx2-WT mice liver after 2/3 PHx. The total RNAs were extracted from Zhx2-KO^{hep} and Zhx2-WT mice liver at 48 h after 2/3 PHx for RNA sequencing. Then the data were used for gene set enrichment analysis.

It is a very good suggestion that using sub-clinical GWAS hits to provide evidence at genetic level for a clinically relevant involvement of ZHX2 in liver disease in human. As suggested, we have searched for indications from publicly available data. Unfortunately, we did not find any GWAS data that directly shows involvement of ZHX2 in liver disease in human. But we do find two GWAS studies show that ZHX2 is relevant to cardiovascular disease, a lipid metabolism deregulation disease, in human (**Bis et al., Nat Genet 2010, PMID: 21909108**; **Li et al., Gene 2015, PMID: 25746325**). Lipid metabolism dysregulation also contributes to NAFLD, NASH and T2D, indicating ZHX2 might play a role in liver diseases in human.

Assuming the above point is proven, the proposed model of how Zhx2 antagonizes ETC OXPHOS is currently inconsistent. In Figure 5, authors propose that OXPHOS suppression is due to the activity of ZHX2 as a transcriptional repressor, occupying genes in the ETC. These data are not so convincing due to the small number of genes found to be direct targets. However in Figure 6, authors switch to a different mechanism, where Zhx2 appears to potentiate the expression of genes involved in ubiquitinylation UPS system, including FBXW7, leading to increased PGC1a ubiquitinylation and degradation. In this respect, it is unclear if Zhx2 is acting directly as a transcriptional activator or via other mechanisms, and if so, what other mechanisms that are downstream of its earlier proposed role as a transcriptional repressor.

Response: Although ZHX2 was firstly identified as a transcriptional repressor (**Kawata et al., Biochem J 2003, PMID: 12741956**), recent research showed that ZHX2 could act as a transcriptional activator at certain circumstance (**Zhang et al., Science 2018, PMID: 30026228**). In this manuscript, we defined two mechanisms by which ZHX2 control mitochondrial OXPHOS. On the one hand, ZHX2 transcriptionally repressed expression of mitochondrial ETC genes *via* binding to their promoters. Based on our RNA-seq data, we did see that lot of genes in mitochondrial OXPHOS pathway are increased in hepatocytes

from *Zhx2*-KO^{hep} mice 48 h after 2/3 PHx (**Fig. 5A**). These results were confirmed in mice hepatocytes and human hepatic organoids after *Zhx2* knockout or knockdown (**Fig. 5C and Supplementary Fig.5B**). Notably, EMSA, luciferase reporter assay and pull-down assay validated the direct binding of ZHX2 to the specific motif on promoters of these 6 genes (**Fig.5G, 5H and Supplementary Fig.5F-5H**). On the other hand, ZHX2 negatively regulated mitochondrial biogenesis in PGC-1 α dependent manner. Gene co-expression network analysis showed PGC-1 α in the central node of the ETC gene network in hepatocytes from *Zhx2*-KO^{hep} mice, and ZHX2 changed protein level but not mRNA level of PGC-1 α (**Fig.6A, 6B and Supplementary Fig.6A-6C**). In mechanism, the protein and mRNA levels of FBXW7, an E3 ligase of PGC-1 α , were decreased in ZHX2 knockout or knockdown cells (**Fig.6F, 6G and Supplementary Fig.6F-6G**). Further investigation determined that ZHX2 promoted PGC-1 α ubiquitination and degradation *via* FBXW7 (**Fig.6H**). To decipher how ZHX2 regulates FBXW7 expression, we did the following experiments. We further analyzed our ChIP-seq data identified one of ZHX2-binding motif located on the promoter of FBXW7 (-1045~-1038 bp). Then we cloned FBXW7 promoter region containing ZHX2-binding motif to construct the luciferase reporter vector. The luciferase reporter assays showed that ZHX2 transfection increased luciferase activity compared to control-transfected cells. Importantly, ChIP assays demonstrated the enrichment of ZHX2 on FBXW7 promoter (**Supplementary Fig.6H-6J**) (**page 14, line 317-319**). All these data suggest that ZHX2 transcriptionally activates FBXW7. Previous reports that ZHX2 works as a transcription factor by interaction with NF-YA or RelA/p65 to repress or promote gene expression (**Kawata et al., Biochem J 2003, PMID: 12741956; Zhang et al., Science 2018, PMID: 30026228**). Therefore, ZHX2, by interacting with different partners, represses or promotes gene expression in a context dependent manner. We had updated these new data and discussed these points in the revised manuscript (**page 18, line 407-411**).

In figure 4, the use of Metformin and FCCP as methods to demonstrate that Zhx2 is acting on OXPHOS to affect liver regeneration are misleading. Metformin has pleiotropic effects, including activation of AMPK which should lead to a general augmentation of OXPHOS

biogenesis in addition to the specific inhibition of Complex I. FCCP administration causes depolarization and is generally toxic. To conclude that FCCP reverses the regenerative effect Zhx2 KO settings, thereby proving that Zhx2 acts via OXPHOs regulation, is really misleading. Unless the authors include a negative control whereby FCCP administration does not affect the regenerative effects of another gene (not related to OXPHOS), I would interpret the findings in Figure 4 to be caused by nonspecific toxicity of FCCP.

Response: We do understand the concern of reviewer and agree with that FCCP had nonspecific toxicity and metformin is not OXPHOS specific inhibitor. As our knowledge, currently there is no find a drug or technique which specifically targets on mitochondrial OXPHOS but has no effect on other liver regeneration pathways or regulators. That's the reason we used alternative methods to verify the role of mitochondrial OXPHOS in ZHX2-mediated regulation of liver regeneration. Metformin has been widely used clinically because of its desirable safety profile (**Knowler et al., N Engl J Med 2002, PMID: 11832527**) and has been believed to disrupt mitochondrial function by targeting mitochondrial complex I (**Wheaton et al., Elife 2014, PMID: 24843020**) or by inhibiting glycerol phosphate dehydrogenase to alter the mitochondrial ETC in liver cells (**Madiraju et al., Nature 2014, PMID: 24847880**). Our data showed that metformin treatment abolished the effect of *Zhx2* deficiency on promoting mitochondrial ATP generation and improving liver regeneration (**Fig. 4**). To exclude the potential non-specific effect of metformin, we included FCCP, an uncoupler of mitochondrial oxidative phosphorylation, to inhibit mitochondrial OXPHOS during liver regeneration. FCCP also has been widely used in the experimental research investigating the roles of mitochondria in cellular function (**Demine et al., Cells 2019, PMID: 31366145; Wang et al., Sensors 2021, PMID: 33923058**). Our results demonstrated that FCCP treatment also eliminates the effect of *Zhx2* deficiency on promoting mitochondrial ATP generation and improving liver regeneration (**Supplementary Fig.4**). Specifically, we also included PGC-1 α inhibitor, PGC-1 α siRNA and overexpression of PGC-1 α both *in vitro* and *in vivo* studies to verify ZHX2 regulates liver regeneration through PGC-1 α mediated-OXPHOS (**Fig.7 and Supplementary Fig.7**). Although we could not exclude all the no-specific effects of these methods, results from these could give us the indications for getting the conclusion.

Specific comments:

- Textual comments:

o Grammatical and syntax errors throughout the manuscript that need to be fixed, especially in the introduction.

Response: We are sorry for these mistakes. Our collaborator, Nailin Li (Karolinska Institute, Stockholm, Sweden) had asked his colleagues to help us correcting these grammatical and syntax errors. All the authors also had carefully gone through the revised manuscript.

o Descriptions of “ectopic expression”, “augmented expression” etc, is difficult to follow if ZHX2 in the system is overexpressed. Best to keep to one definition of overexpression.

Response: Thanks for comments. As suggested, we had changed these words to “overexpression” in the revised manuscript (**page 4, line 89; page 6, line 131,138; page 11, line 254; page 14, line 304, 308; page 15, line 329 and so on**).

- Figure 2: Chip-SEQ data presented are restricted to OXPHOS genes. What about other classes of genes that are affected in Zhx2 KO that impinge on hepatocyte proliferation? Does Zhx2 directly affect transcription of cyclin A2/B1 etc and increase cell proliferation? Do they have Zhx2 binding sites? This is also observed in Fig S3C where cell cycle genes are upregulated in Zhx2-KO mice. Figure 2D: data in plot is reverse compared to other panels.

Response: As we are focus on the role of ZHX2 in regulating mitochondrial OXPHOS, we only presented genes involving in mitochondrial OXPHOS and examined whether ZHX2 transcriptionally suppresses 6 ETC genes identified by RNA-seq and ChIP-seq data in the main manuscript. In the supplement data, we also presented that ZHX2 binds to the promoter region of some previously reported targets of ZHX2, including cell cycle gene cyclin E (**Supplementary Fig.5F**). Although our ChIP-seq did not find direct ZHX2-binding motif in cyclin A2, previous study demonstrated that there is a NF-YA-binding motif in the promoter region of cyclin A2 (**Kramer et al., Cancer Res 1997, PMID: 9371512**). It has been reported that ZHX2 interacts with NF-YA and regulating NF-YA target (**Kawata et al.,**

Biochem J 2003, PMID: 12741956). Thus, ZHX2 might transcriptionally repress cyclin A2 via NF-YA. In addition, as suggested, we had corrected the bar graph of *Cyclin D1* in the revised Fig. 2D.

- Figure 3:

o Figures and legend do not correspond to each other.

Response: Sorry for this inconsistency. We had corrected the figure legends of **Fig. 3I and 3J** in the revised manuscript (**page 33, line 846-852**).

o What are the differences at baseline for control vs Zhx2 KO without injury? Are these pathways enriched even further upon injury?

Response: To evaluate the differences of *Zhx2*-KO^{hep} mice and control mice at baseline, RNA sequencing was performed with hepatocytes isolated from *Zhx2*-KO^{hep} mice and control mice without PHx. Interestingly, GSEA did show the enrichment of mitochondrial OXPHOS pathways in *Zhx2*-KO^{hep} mice liver, while the normalized enrichment score, *p* value and *q* value were not as significant as those in the mice at 48 h after 2/3 PHx (**Fig. 3A and Reviewer only Figure 5**). All these data demonstrated that mitochondrial pathways enriched even further upon injury. Raw and processed sequencing are deposited to National Center for Biotechnology Information Sequence Read Archive (PRJNA989038).

Reviewer only Figure 5. GSEA was used to analyze the genes with differential mRNA levels. The total RNAs were extracted from *Zhx2*-KO^{hep} and *Zhx2*-WT mice liver for RNA sequencing. Then the data were used for gene set enrichment analysis.

o Fig 3A-B: Gene sets are very general pathways. Are there specific mitochondrial pathways that are enriched? Fig 3B: "Energy metabolism" is as enriched as other pathways (lipid, amino acid metabolism).

Response: We did find the enrichment of specific mitochondrial pathways with the RNA-seq data, such as OXPHOS and mitochondrial showed in Fig.3A. We emphasized this in the revised manuscript (**page 8, line 192-193**). As Fig.3B is concerned, several pathways were as enriched as “Energy metabolism” such as “Lipid metabolism”, “Amino acid metabolism”, both of which are related with energy. In the revised manuscript, we had described the enrichment of lipid and amino acid metabolism (**page 9, line 198-200**).

o Figure S3D: fold change >1.2 seems very lenient, are there pathways enriched when cut off is more stringent?

Response: Due to the compressive effect of labeling in quantitative protein labeling (Nuwaysir *et al.*, *J Am Soc Mass Spectrom* 1993, PMID: 24227670; Cheng *et al.*, *Methods Mol Biol* 2016, PMID: 26584918), and after referring to the literature (Shi *et al.*, *Cell Metab* 2021, PMID: 33440166), we used FC>1.2 as the threshold condition for screening differential proteins.

o Results line 228: expression correlative and only a small number of genes (4) were shown to be direct repression targets. As such, authors should be careful to conclude ZHX2 as a direct inhibitor of OXPHOS.

Response: Thanks for these very useful comments. In order to accurately describe the data, we had changed the sentence to “indicating that ZHX2 might regulate mitochondrial OXPHOS in liver diseases” in the revised manuscript (**page 10, line 216-217**).

- Figure 4 :

o Figure 4A: Metformin can already disrupt membrane potential. JC-1 might not be the best readout for mitochondrial health here.

Response: Thanks for this constructive comment. We have used TMRM to indicate mitochondrial health in the revised manuscript. The results showed that knockdown of ZHX2 increased TMRM intensity, metformin treatment abolished ZHX2 knockdown-induced increasing of TMRM intensity (**Fig. 4A**) (**page 33, line 857**).

o Figure 4D/4E/4F/4G: *The observed increases in gene expression are quite small, albeit not statistically significant in KO upon metformin treatment. Are there other mitochondria-independent pathways that help recovery?*

Response: Fig.4D,4E,4G,4H showed the liver weight/body weight ratio, cell cycle related genes levels, cell proliferation and ATP levels respectively. Typically, mice liver weights fall in the range of 3-5% body weight. After 2/3 PHx, liver weight/body weight ratio is around 2%. Thus, our results showed the changes of liver weight/body weight ratios within the range. Importantly, deficiency of *Zhx2* increased the liver weight/body weight ratio and this enhancement was abolished by metformin treatment, which is consistent with our hypothesis that *Zhx2* inhibits liver regeneration via OXPHOS. While in **Fig.4E and F**, levels of cell cycle related genes and BrdU (marker of cell proliferation) in *Zhx2*-KO^{hep} mice increased by 50% to 100% (from 10% to 20% at 36h, and from 20% to around 35% at 48h) which are clear changes. Also, these differences were almost disappeared after metformin treatment. As Fig.4G is concern, ATP concentrations are maintained with a very narrow range normally. While ATP/AMP ratio is the clear changed parameter compare with ATP concentration (**Miguel Beato, Nat Rev Mol Cell Biol 2019, PMID: 31175344**). Therefore, we had measured both parameters to show that knockout of *Zhx2* promoted ATP generation in mice liver cell after 2/3 PHx, and these phenomena were eliminated by metformin treatment (**Fig. 4G**). All the results are statistically significant and credible. Still, we could not exclude the potential involvement of other pathways in liver regeneration which is beyond the scope of this present manuscript.

- Figure 5B: *What about other ETC proteins? Are other mitochondrial pathways also affected since mitochondrial mass increases?*

Response: Fig.5B list all the differential expressed mitochondrial ETC proteins in our proteomic data. Since the intensity of sequencing will lead to miss some useful information, we performed western blot using total OXPHOS Rodent WB Antibody Cocktail (**Abcam, ab110413**), which was widely used to detect mitochondria by western blot (**Lei Huang et al., Adv Sci 2022, PMID: 34747141**). The results showed that the protein levels of ATP5A, MTCO1, SDHB and NDUF8 were higher in *Zhx2*-KO^{hep} mice livers than those in *Zhx2*-WT

mice livers (**Reviewer Only Figure 6A**). In addition, we did further analysis with the proteomic data and found some other differential expressed mitochondrial proteins locating in mitochondrial outer membrane (MOM), mitochondrial inner membrane (MIM), mitochondrial matrix (Matrix) and mitochondrial intermembrane space (IMS). The data showed that these mitochondrial locating proteins increased in *Zhx2*-KO^{hep} mice livers, consisting with our conclusion that ZHX2 repress mitochondrial mass. We have updated these data as **Supplementary Fig.5A** in revised manuscript (**page 11, line 251-252**). And, further analysis of RNA-seq data showed that mitochondrial pathways, mitochondrial calcium ion transport and mitochondrial fatty acid beta oxidation, also enriched in *Zhx2*-KO^{hep} mice liver at 48 h after 2/3 PHx (**Reviewer Only Figure 6B**).

Reviewer only Figure 6. ZHX2 is negatively associated with mitochondrial function in mice liver after 2/3 PHx. (A) The protein levels of ETC proteins were determined in *Zhx2*-KO^{hep} and *Zhx2*-WT mice at 48 h after 2/3 PHx using combination of antibodies by western blot. **(B)** GSEA were used to analyze RNA-seq data from *Zhx2*-KO^{hep} and *Zhx2*-WT mice liver at 48 h after 2/3 PHx. Two mitochondrial pathways were presented.

- Figure 6D: *overexpression levels not compatible to controls, especially in CQ panel.*

Response: Thanks for this carefully review. This experiment has been repeated at least 3 times with consistent results. To further confirm, we had performed this experiment again and showed the new image in the revised **Fig.6D** in which the overexpression levels, especially in CQ panel, were compatible to controls.

- Figure 7D: *only a few ETC genes are affected by Zhx2 direct binding, and a few are affected by PGC1a instability. Are the reduction in expression of these genes enough for reduction of mitochondrial mass?*

Response: Emerging research show that variant of one ETC gene could change the mitochondrial functions (**Choi et al., Nat Neurosci 2019, PMID: 31086314; Olahova et al., Am J Hum Genet 2018, PMID: 29478781; Schuelke et al., Nat Genet 1999, PMID: 10080174**). For example, Zou reported that knockdown of *Ndufs1*, one of the core subunits of mitochondrial complex I, decreased the mitochondrial DNA content, mitochondrial membrane potential (MMP), and mitochondrial mass (**Zou et al., Oxid Med Cell Longev 2021, PMID: 33763166**). Lee showed that knockdown of *Sdha*, a component of mitochondrial complex II, also reduced mitochondrial mass in human CB-CD34⁺ cells (**Lee et al., Stem Cell Reports 2016, PMID: 27346679**). Based on these findings, we speculate that reducing of several ETC genes could decrease mitochondrial mass. Furthermore, it is well known that PGC-1 α is a master regulator of mitochondrial biogenesis (**Wenz T, Mitochondrion 2013, PMID: 23347985**). A large body of research demonstrated that decreased level of PGC-1 α leads to reduced mitochondrial mass (**S Baldelli et al., Cell Death Dis 2018, PMID: 25375380; Joseph T Rodgers et al., Nature 2021, PMID: 15744310; Marie Lagouge et al., Cell 2006, PMID: 17112576**), which is consistent with our data.

- Figure 8D: Is Tom20 PGC1a controlled? Are there other mitochondrial markers apart from TOM20 that are coexpressed with Zhx2 upon DILI?

Response: Thanks for the comments. Although TOM20 is not directly controlled by PGC-1 α , it is widely used as the marker of mitochondrial. As suggested, we had used another mitochondrial marker COX IV to stain the DILI samples. The results showed that high levels of ZHX2 is associated with low levels of COX IV (**Supplementary Fig.7F**). We had updated these new data in revised manuscript (**page 16, line 365-367**).

Reviewers' comments:

Reviewer #1 (Remarks to the Author):

The authors have addressed satisfactorily the concerns raised in the original submission.

Reviewer #2 (Remarks to the Author):

The authors have satisfactorily responded to all my questions.

Reviewer #3 (Remarks to the Author):

I appreciate the efforts of the authors in responding to my concerns, but I remain unconvinced that *Zhx2* deletion provides any long-term improvements in liver regeneration following injury without further examination of the long-term effects of *Zhx2* deletion on liver function following injury.

For instance, in NASH (PMID :34545586), *Zhx2* deletion worsens NASH progression. This is at odds with the current study that *Zhx2* deletion improves liver regeneration. Enhancing mitochondrial OXPHOS (the proposed mechanism in this study) should be protective in NASH (PMID: 36057633), but yet it isn't. Similarly, the data showed in Reviewer Only Figure 3 is at odds with the conclusions that *Zhx2* ablation promotes liver regeneration, since there is increased liver fibrosis. So while there are some observed increases in cell proliferation and reduced serum ALT in the first 24 hours after injury, I am still not convinced that this is biologically significant.

Response to Reviewer 3#:

*I appreciate the efforts of the authors in responding to my concerns, but I remain unconvinced that *Zhx2* deletion provides any long-term improvements in liver regeneration following injury without further examination of the long-term effects of *Zhx2* deletion on liver function following injury.*

Thanks for your understanding of our efforts and your kind comments. These will help us improve our paper a lot. We will try our best to answer your concern. In fact, we did long-term effects of *Zhx2* deletion on liver function following 2/3 PHx. Although the liver/body weight ratios following 2/3 PHx were recovered faster in *Zhx2*-KO^{hep} mice than that in *Zhx2*-WT mice (**Figure 2A**), the liver/body ratios are comparable in *Zhx2*-KO^{hep} mice and WT mice at 7days after PHx (**Figure 2A**) and maintained stable even after one month. In accordance, serum levels of ALT and AST were comparable between *Zhx2*-KO^{hep} mice and *Zhx2*-WT mice at 7 days after 2/3 PHx, which were similar with that in mice at homeostatic condition (**Supplementary Figure 2E**), suggesting that *Zhx2* deficiency protect liver from acute injury for long time. This was further supported by H&E staining of liver tissues which showed no difference between *Zhx2*-KO^{hep} mice and WT mice (**Supplementary Figure 2F**). These data consistent with the mice at homeostatic conditions. Dr. Brett T Spear' group reported that manipulation of *Zhx2* affects expression of genes, such as *AFP*, *Mup*, and *Cyp*, but has no visible effects on mice liver even feeding on normal conditions for 9 months (**Gene Expr, PMID: 27197076; Hepatol Commun, PMID: 36194180; J Biol Chem, PMID: 28258223**). Consistently, we detected no difference

in liver between *Zhx2*-KO^{hep} mice and WT mice after 6 months feeding at normal conditions. As shown in **Reviewer Only Figure 1**, the liver/body weight ratios, serum levels of ALT and AST, and liver tissue structural had no significant difference between *Zhx2*-KO^{hep} mice and WT mice. These data suggest that the effects of *Zhx2* deficiency is dependent on the challenge using in the model.

Reviewer only Figure 1. The comparison of *Zhx2*-KO^{hep} mice and *Zhx2*-WT mice liver after feeding for 6 months at normal conditions. (A) Liver/body weight ratios of *Zhx2*-WT and *Zhx2*-KO^{hep} mice were displayed. **(B)** The serum ALT and AST levels were detected in *Zhx2*-WT and *Zhx2*-KO^{hep} mice. **(C)** Representative images of H&E staining for liver sections from *Zhx2*-WT and *Zhx2*-KO^{hep} mice were displayed.

Although our data fully supported that ZHX2 regulates OXPHOS to control liver recovery from acute injury, we do agree with you that our current manuscript did not provide strong evidence for the long-term effects of ZHX2-mediated OXPHOS regulation in metabolic liver diseases. To make our description more clearly and acutely, we have adjusted our revised manuscript according your comments as following: First, in order to accurately present our findings, we have changed our titles to “**ZHX2 emerges as a negative regulator of mitochondrial oxidative phosphorylation during acute liver injury**” (Page 1). Second,

in the revised introduction, we focused on describing the progress about mitochondrial function in acute liver injuries and modulation of mitochondrial regulator in recovery of liver after acute injuries (**Page 3, Paragraph 1; Page 4, Paragraph 1**). Third, in the revised discussion, we mainly discussed effect of ZHX2 in repair of liver after acute injuries and the difference of ZHX2 effects and mitochondrial function between acute and long-term injuries (**Page 17, Paragraph 2 and 3; Page 18; Page 19, Paragraph 1**). Finally, we replaced the online data analysis showing enrichment of OXPHOS pathway in ZHX2 low HCC/NAFLD cohort with the analysis using acute liver failure cohort (**Page 5, Paragraph 2**), and deleted the public published data analysis showing the negative association of ETC genes and ZHX2 in patients with HCC and NAFLD (**original Fig.S3F**).

For instance, in NASH (PMID :34545586), Zhx2 deletion worsens NASH progression. This is at odds with the current study that Zhx2 deletion improves liver regeneration. Enhancing mitochondrial OXPHOS (the proposed mechanism in this study) should be protective in NASH (PMID: 36057633), but yet it isn't. Similarly, the data showed in Reviewer Only Figure 3 is at odds with the conclusions that Zhx2 ablation promotes liver regeneration, since there is increased liver fibrosis. So while there are some observed increases in cell proliferation and reduced serum ALT in the first 24 hours after injury, I am still not convinced that this is biologically significant.

Thanks for your careful review of our work. We are glad to discuss our new findings and

research about this field with you. In this manuscript, our data clearly verified that deficiency of *Zhx2* accelerated liver recovery after acute injuries. Indeed, the effect of *Zhx2* in long-term liver injuries is also very interesting. In the first round of revision, we just simply presented previous reports and our ongoing data about the effects of *Zhx2* in long-term liver injuries which seem odds with the function of *Zhx2* in acute liver injuries. In order to eliminate your concerns, we discussed these results in detail as following:

First, as pointed in Science paper (**Science**, PMID: 30026228), ZHX2 affects different signals to exert its function in different models at certain conditions. This was further supported by studies reporting different roles of ZHX2 in liver diseases. For instance, ZHX2 overexpression suppresses HCC proliferation both *in vitro* and *in vivo* (**EBioMedicine**, PMID: 32114388; **Gastroenterology**, PMID: 22406477), while ZHX2 is required for diethylnitrosamine-induced liver tumor formation in C57BL/6 mice (**Hepato Commu**, PMID: 36194180). *Zhx2* deletion promotes NAFLD and NASH progression in HFHC- and HFD-treated mice (**Hepatology**, PMID: 34545586; **Cell Death Differ**, PMID: 31740790). Here, we found that deficiency of *Zhx2* accelerates liver recovery by enhancing mitochondrial OXPHOS in 2/3 PHx and acute CCl₄-treated mice models. The seeming inconsistency of our data in acute liver injury with literature on NASH might be due to the ZHX2-mediated context dependent manner. In fact, the treatments and operations in these models are different that will cause the effects of ZHX2 are different. And, the dominant ZHX2-regulated signals also might be varied in these models upon stimulation.

Second, ZHX2-regulated signals also displays a context dependent manner. As reference

(Hepatology, PMID: 34545586) reported, loss of *Zhx2* damagingly activates mTOR signaling in HFHC-treated mice liver. Here, our RNA sequencing data showed that mitochondrial OXPHOS is the most significantly changed signal in mice liver 48 h after 2/3 PHx. As you suggested, we also examined this signal in mice liver under hemostatic state. The results showed that although mitochondrial OXPHOS were enriched in *Zhx2*-KO mice liver, its normalized enrichment score and *p* value were not as significant as those in the mice at 48 h after 2/3 PHx. Consistently, the ATP levels in *Zhx2*-WT and *Zhx2*-KO mice liver had no significant difference after sham operation (**Reviewer Only Figure 2**). This supports the hypothesis that ZHX2 regulates different signals in a context dependent manner. These might also explain the seeming inconsistency between acute injury (showing *Zhx2* deletion promotes liver regeneration) and long-term injury (reporting *Zhx2* deletion worsens NASH in HFHC-treatment). Besides, our ongoing liver fibrosis data showed that *Zhx2* deletion increased liver fibrosis, which is due to the role of *Zhx2* in regulating hepatic redox balance after long time low-dose CCl₄ treatment (**Reviewer Only Figure 3**). Again, this further supported that ZHX2 takes effects to activate signals in a context dependent manner. Above all, the biological functions of ZHX2 in liver are very complex which need further investigation.

Reviewer only Figure 2. Mitochondrial activity was examined in *Zhx2-KO^{hep}* and *Zhx2-WT* mice.

The total RNAs were extracted from *Zhx2-KO^{hep}* and *Zhx2-WT* mice liver for RNA sequencing. GSEA was performed using the data from the mice liver 48 h after 2/3 PHx (A) and sham operation (B). (C) ATP levels were determined in *Zhx2-KO^{hep}* and *Zhx2-WT* mice liver after sham operation.

Reviewer only Figure 3. The role of ZHX2 in CCl₄-induced liver fibrosis. 25% CCl₄ (5 mL/kg) were used to treat *Zhx2-KO^{hep}* and *Zhx2-WT* mice for two months. (A) ZHX2 protein levels were detected in

Zhx2-WT and *Zhx2*-KO^{hep} mice liver tissues by western blot. **(B)** Sirius red staining under the light microscope indicates total collagen deposits. Representative images and quantitative data were presented. **(C)** Masson staining for collagen in liver biopsy specimens from CCl₄-treated mice. Representative images and quantitative data were presented. **(D)** GSEA showed that reactive oxygen species is the significantly change pathways in *Zhx2*-deleted mice fibrotic liver. **(E)** The ROS levels were detected in *Zhx2*-KO^{hep} and *Zhx2*-WT mice after inducing liver fibrosis using flow cytometry.

Third, mitochondrial oxidative capacity varies broadly across the spectrum of obesity and NAFLD, and the contribution of alterations of mitochondrial function to NASH progression is controversial. Many papers showed that enhancing of mitochondrial OXPHOS alleviated NASH progression. As the reviewer listed the paper that published on nature communications (**Nat Commun, PMID: 36057633**). Emerging research demonstrated that loss of mitochondrial OXPHOS could reduce NASH progression. For example, Hepatocyte-specific deletion of AIF indicated that the loss of mitochondrial OXPHOS protected against diet induced steatosis and NASH progression (**Cell, PMID: 17981116**). Therefore, ablation of *Zhx2* enhancing mitochondrial OXPHOS might also has the possibility to promote NASH progression. Together, mitochondrial dysfunction in humans is manifest as a variety of disorders with clinical outcomes largely dependent on the magnitude and tissue distribution of the impairment (**Nature, PMID: 16572142; Science, PMID: 10066162**). Thus, the function of mitochondrial function in liver diseases is a very interesting topic.

In addition, by searching literatures, we found that many genes represented opposite phenomena in acute and long-term liver injuries. For instance, the AP-1 transcription factor

c-Jun is a key regulator of hepatocyte proliferation. Mice lacking c-Jun in the liver displayed impaired liver regeneration after PHx (**Genes Dev, PMID: 16912279**). Meanwhile, c-Jun/AP-1 promotes liver fibrosis during non-alcoholic steatohepatitis (**Cell Death Differ, PMID: 30778201**). Follistatin-like protein 1 (FSTL1) is widely recognized as a secreted glycoprotein. FSTL1 is significantly elevated during active liver regeneration (**Cancer Res, PMID: 34551961**). Rao *et al.* showed that FSTL1 promotes liver fibrosis by reprogramming macrophage function (**Gut, PMID: 35140065**). Therefore, the same gene could activate differently signals to exert opposite functions in the progression of liver diseases.

Again, we thank for your insightful comments. We try our best to eliminate your concern.

We do agree with you that in the current manuscript we did not provide strong evidence for the long-term effects of ZHX2-mediated OXPHOS regulation in metabolic liver diseases.

In these manuscript, we mainly focused on effects of ZHX2-mediated OXPHOS regulation in recovery of mice liver from acute injuries. Therefore, we have adjusted our revised manuscript as describing in answer your first part comments. Since effect of ZHX2 on liver fibrosis is our ongoing work, we suggest that could be a follow-up paper. Above all, these comments really improves our work a lot. Hopefully, these changes according your comments will eliminate your concerns. Importantly, as you point out, these paper raises an interesting finding of potential interest to a wide audience that *Zhx2* is a modifier in the ability of liver to regenerate following acute injuries.

REVIEWERS' COMMENTS

Reviewer #3 (Remarks to the Author):

The authors and sufficiently addressed my concerns and I thank them for a careful review of the literature and for presenting the new data. This has improved the manuscript substantially.